# MEMORY-EFFICIENT LLM PRETRAINING VIA MINIMALIST OPTIMIZER DESIGN

## ABSTRACT

Training large language models (LLMs) typically relies on adaptive optimizers such as Adam, which introduce extra operations and require significant more memory to maintain first- and second-order moments than SGD. While recent works such as GaLore, Fira and APOLLO have proposed state-compressed variants to reduce memory consumption, a fundamental question remains: *What are the minimum modifications to plain SGD needed to match state-of-the-art pretraining performance?* We systematically investigate this question using a bottom-up approach, and identify two simple yet highly (memory- and compute-) efficient techniques: (1) column-wise gradient normalization (normalizing the gradient along the output dimension), which boosts SGD performance without momentum; and (2) applying first-order momentum only to the output layer, where gradient variance is highest. Combining these two techniques lead to SCALE (Stochastic Column-normAlized Last-layer momEntum), a simple optimizer for memory efficient pretraining. Across multiple LLaMA models (60M–1B), SCALE matches or exceeds the performance of Adam while using only 35–45% of the total memory. It also consistently outperforms memory-efficient optimizers such as GaLore, Fira and APOLLO, making it a strong candidate for large-scale pretraining under memory constraints. For LLaMA 7B model, SCALE outperforms the state-of-the-art memory-efficient methods APOLLO and Muon, in terms of both perplexity and memory consumption.

## 1 INTRODUCTION

Adaptive optimizers such as RMSProp (Hinton et al., 2012), Adam (Kingma & Ba, 2015) are the default optimizers for large-scale attention-based deep neural networks, particularly large language models (LLMs). While effective, these optimizers incur significant memory overhead due to their reliance on maintaining both first- and second-moment estimates of the gradient, which are also known as *optimizer states*. Specifically, Adam requires storing two additional gradient states per parameter, tripling the memory usage compared to vanilla stochastic gradient descent (SGD).

On the other hand, despite its superior memory efficiency, the vanilla SGD performs poorly when applied directly to LLM training, due to the absence of adaptive scaling in the update step (See Figure 2 for experiment results, also see Zhao et al. (2025); Zhang et al. (2020b)). This has motivated a wave of recent research focused on developing memory-efficient alternatives to Adam that aim to retain its performance while reducing memory consumption. These include compression-based algorithms such as GaLore (Zhao et al., 2024), Fira Chen et al. (2024) and APOLLO Zhu et al. (2025). Novel optimizers, such as Muon (Jordan et al., 2024), Scion (Pethick et al., 2025) and SWAN (Ma et al., 2024), also by design require less memory than Adam. These approaches often introduce new algorithmic components—such as different forms of gradient normalization, whitening and rescaling, momentum variants—and combine them in various ways.

However, despite the growing literature on memory-efficient optimizers, there has been no systematic study to identify which specific algorithmic components or techniques are most essential for designing highly performant yet minimal-memory optimizers. For instance, are both first- and second-order momentum terms strictly necessary for effective training? Among the many forms of gradient normalization, which subset of them strike the best trade-off between performance, memory, and computational cost? In the absence of such a principled investigation, it remains unclear how to balance optimizer complexity against memory savings.

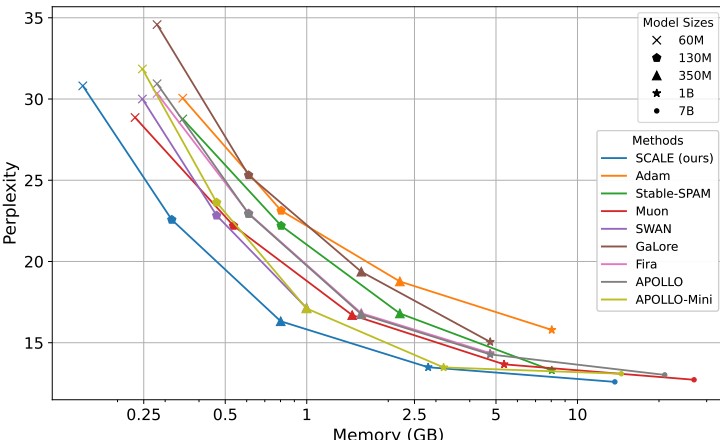

Figure 1: Perplexity v.s. memory consumption among a number of SOTA algorithms. Solutions achieved towards the left-bottom side of the plot represent better performance/memory trade-off (see Appendix A.2 for the details of the memory estimation).

This motivates our central research question:

> Can we design a memory efficient optimizer with minimum modifications to plain SGD that achieves state-of-the-art pretraining performance?

In this work, we address this question through a *bottom-up*, *minimalist* approach. Rather than starting from existing adaptive optimizers, we systematically evaluate the role of fundamental components, namely normalization and momentum, to determine the smallest set of techniques needed to bridge the gap between vanilla SGD and Adam, with a memory efficiency focus. Towards this end, we perform extensive experiments to identify key components (such as different forms of normalization and different levels of momentum) that can effectively enhance the performance of vanilla SGD with minimum memory overhead. We then justify the design choices using a combination of theoretical insights and empirical evidence.

**Contributions.** Our study suggests that two techniques, when used together, are particularly effective: *(i)* Among various gradient normalization techniques in the literature, column-wise gradient normalization (normalizing along the output dimension) can significantly boost SGD performance, while having simple closed-form solution and with no extra memory required; and *(ii)* adding first-order momentum exclusively to the output layer, where gradient variance is highest (see Figure 4), is surprisingly effective to further boost training performance with minimum memory overhead. Combining these two techniques lead to Stochastic Column-normalized Last-layer momentum (SCALE), a memory efficient optimizer which requires roughly the same amount of memory as vanilla SGD. For example, for 1B (resp. 7B) model, SCALE only requires 10% (resp. 2%) more memory as compared to vanilla SGD. Meanwhile, SCALE achieves similar performance as compared with SOTA optimizers Adam and Muon, with only 35% and 52% of the memory cost for training 1B models, respectively. Comparing to other memory efficient optimizers, SCALE achieves superior performance to GaLore, Fira and Apollo, with only 59% of the memory cost for training 1B models. See Figure 1 for an illustration of performance and memory trade-off among different SOTA algorithms.

## 1.1 RELATED WORKS

**Memory-efficient variations of Adam.** A recent line of works aims to improve the memory efficiency of Adam by compressing the historical states stored, namely the first- and second-order statistics. Early works include Adafactor (Shazeer & Stern, 2018), which estimates that optimizer states using per-row and per-column moving averages, SM3 (Anil et al., 2019), a memory efficient AdaGrad via grouping second-order momentum, and CAME (Luo et al., 2023), which improves upon Adafactor via matrix factorization. A number of recent works use gradient projections to compress the optimizer states. GaLore (Zhao et al., 2024), being one of the pioneering works, stores the states in a low-rank subspaces that captures most gradient information. Fira (Chen et al., 2024) achieves superior performance than GaLore by re-introducing full-rank information to the low-rank gradients. APOLLO (Zhu et al., 2025) constructs the update based on gradient scaling factors that

are estimated from the ratio between the low-dimensional gradient and the low-dimensional Adam update; APOLLO-Mini is a memory efficient version of APOLLO by estimating in a rank-1 subspace. GRASS (Muhamed et al., 2024) improves GaLore by designing sparse projection matrices guided by the norm of the rows of the gradients. SlimAdam (Kalra et al., 2025) compresses second-order moments based on signal-to-noise analysis. Some other methods group parameters into blocks and apply block-wise updates (Luo et al., 2024; Ramesh et al., 2024; Pan et al., 2024) or block-wise scaling (Zhang et al., 2024) to further reduce the memory costs.

**Towards removing optimizer states.** More recent works start to directly remove the optimizer states as required by Adam. Muon (Jordan et al., 2024; Liu et al., 2025) has shown impressive results using only first order momentum and orthogonalizing it via an iterative algorithm. Scion (Pethick et al., 2025) explores various (layer-wise) normalization schemes for better training performance. SWAN (Ma et al., 2024) and its variant (Scetbon et al., 2025) combine two normalizations (see Section 2 for details) directly to the gradient, matching the performance of Adam (when applied only to the intermediate layers while the first and last layers still use Adam. See Section 4 for details). Zhao et al. (2025) demonstrates SGD with signed momentum is able to give similar performance to Adam. SGD-SaI (Xu et al., 2024) verifies that proper learning-rate scaling at initialization is sufficient to achieve good performance.

## 2 METHODOLOGY

We begin by reviewing several key techniques used in popular optimizers such as Adam and Muon, which have been shown to accelerate pretraining performance for LLMs. This understanding will be critical for our subsequent minimalist design for a memory-efficient optimizer. Denote the optimization problem of the LLM pretraining as

$$\min_{\theta=[\theta_1,...,\theta_L]} \ell(\theta) := \frac{1}{n}\sum_{i=1}^{n}\ell(\theta;\xi_i) \tag{1}$$

where $\ell$ is the loss function, usually taken as the cross entropy loss of predicting the next token, $\theta = [\theta_1,...,\theta_L]$ is the model trainable parameters, with $\theta_l$ the $l$-th layer, $l = 1, 2, ..., L$. With attention-based network, we can simply assume that each $\theta_l \in \mathbb{R}^{d_{l,\text{in}} \times d_{l,\text{out}}}$ is a weight matrix, with the input dimension $d_{l,\text{in}}$ and output dimension $d_{l,\text{out}}$. Here $\xi_i$ with $i = 1, ..., n$ represents training data samples and $n$ is the training data size.

To solve equation 1, **vanilla SGD** draws a small batch of i.i.d samples $\{\xi_{t,b}\}_{b=1,..,B}$ at iteration $t$ and performs update toward the negative stochastic gradient direction:

$$\theta^{t+1} = \theta^t - \eta_t g^t, \quad \text{with} \quad g^t := \frac{1}{B}\sum_{b=1}^{B}\nabla\ell(\theta^t;\xi_{t,b}) \tag{2}$$

where $B$ is batch size and $\eta_t$ is the learning rate. Although memory-efficient, plain SGD performs poorly in LLM training due to the lack of adaptive scaling (Zhao et al., 2025), and we verify this in Figure 2 where we run SGD and Adam on LLaMA 130M pretraining task (see Section 4 for the details of our experiment settings). Therefore we will not include the perplexity result for SGD in the subsequent experiments.

In contrast, adaptive algorithms such as **Adam** (Kingma & Ba, 2015) update parameters using a more sophisticated scheme (where $\odot$ represents element-wise product):

$$\begin{aligned}
m^t &= \beta_1 m^{t-1} + (1-\beta_1)g^t, \\
v^t &= \beta_2 v^{t-1} + (1-\beta_2)g^t \odot g^t, \\
\theta^{t+1} &= \theta^t - \eta_t \frac{m^t}{\sqrt{v^t}+\epsilon}.
\end{aligned} \tag{3}$$

Numerous theoretical works such as Reddi et al. (2019); Zhang et al. (2022b;a) attempted to explain why Adam outperforms SGD for transformers, though consensus remains elusive. A straightforward decomposition of Adam identifies two essential components.

1. **Gradient Normalization**. The key difference between Adam and SGD is the normalization factor $v^t$ in the denominator. One could first normalize each element of SGD, resulting in the

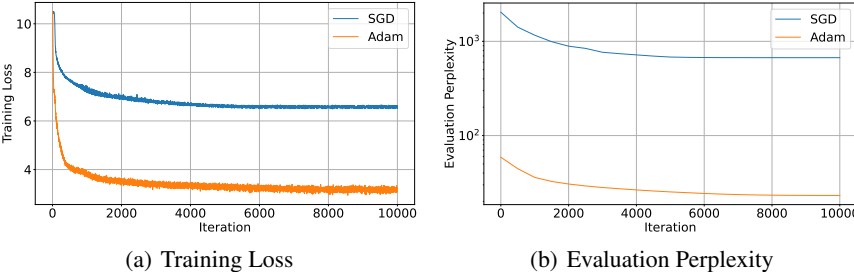

(a) Training Loss           (b) Evaluation Perplexity

Figure 2: Comparison of SGD and Adam training loss and evaluation perplexity curves on LLaMA 130M model. Clearly, SGD is not converging to any reasonable level of perplexity. The Adam and SGD learning rates are 3e-3 and 0.1, respectively. We search with multiple learning rates for SGD, for lower ones the loss decreases even slower and higher ones cause the training to diverge.

sign-SGD update as follows:

$$\theta^{t+1} = \theta^t - \eta_t \frac{g^t}{\sqrt{g^t \odot g^t}} = \theta^t - \eta_t \, \mathrm{sign}(g^t) \tag{4}$$

where $\mathrm{sign}$ stands for taking the sign for each element.

2. **Exponential Moving Average (EMA)**. The stochasticity of the mini-batch sample $\{\xi_{t,b}\}_{b=1,..,B}$ of $g^t$ could be smoothed by taking the exponential moving average (EMA) for both the numerator and denominator for the update equation 4, resulting in the following Adam update of the numerator and denominator:

$$m^t = \beta_1 m^{t-1} + (1 - \beta_1)g^t, \quad v^t = \beta_2 v^{t-1} + (1 - \beta_2)g^t \odot g^t. \tag{5}$$

Combining equation 4 and equation 5 results in the original Adam update equation 3.

This leads to the question: **How much of Adam's benefit arises from gradient normalization versus EMA?** We note that gradient normalization does not require maintaining state, whereas EMA does. Therefore, in the next sections, we will examine each of these components separately and discuss their effectiveness. We will start from the gradient normalization, as the EMA step will introduce extra memory no matter how hard we compress it. Then we will discuss how to use EMA in a more memory-friendly way to further boost the performance of the optimizer.

## 2.1 GRADIENT NORMALIZATION

Gradient normalization is a critical component for large-scale, efficient pretraining, and has remained indispensable even in recent (near)-stateless optimizers (Ma et al., 2024; Huang et al., 2025; Zhu et al., 2025). It is known to accelerate escape from saddle points (Levy, 2016; Murray et al., 2019), improve the effective smoothness of the objective (when the Hessian norm is upper bounded in terms of the gradient norm) (Zhang et al., 2020a; Kunstner et al., 2023), stabilize gradient distributions (Ma et al., 2024), and provide robustness to heavy-tailed noise (Sun et al., 2024).

Denote $G \in \mathbb{R}^{d_{\mathrm{in}} \times d_{\mathrm{out}}}$ the stochastic gradient (we switch to upper letters since we assume the weight blocks $\theta_l$ in equation 1 are matrices). There is a variety of gradient normalization schemes, including

$$
\begin{aligned}
\text{Singular-value normalization:} \quad & UV^\top, \quad \text{where } G = U\Sigma V^\top \text{ (SVD)} \\
\text{Column-wise normalization:} \quad & \left[ \frac{\mathrm{col}_1(G)}{\|\mathrm{col}_1(G)\|}, \dots, \frac{\mathrm{col}_n(G)}{\|\mathrm{col}_n(G)\|} \right] \\
\text{Row-wise normalization:} \quad & \left[ \frac{\mathrm{row}_1(G)^\top}{\|\mathrm{row}_1(G)\|}, \dots, \frac{\mathrm{row}_m(G)^\top}{\|\mathrm{row}_m(G)\|} \right]^\top \\
\text{Sign normalization:} \quad & \mathrm{sign}(G)
\end{aligned}
\tag{6}
$$

Here we assume that $d_{\mathrm{in}}$ is the input dimension and $d_{\mathrm{out}}$ is the output dimension. Therefore row- and column-normalizations correspond to normalizing along the input and output dimensions, respectively. It can be verified that different normalization schemes arise naturally from the steepest ascent direction under different matrix norms (Bernstein & Newhouse, 2024a;b; Pethick et al., 2025; Bernstein & Newhouse, 2025). Many existing optimizers employ different normalization techniques.

For example, Sign-SGD/Adam utilize $\|\cdot\|_{1\to\infty}$ (sign normalization)[1] to achieve gradient normalization, Muon (Jordan et al., 2024) utilizes $\|\cdot\|_{2\to2}$ norm (singular-value normalization), whereas SCION (Pethick et al., 2025) and SlimAdam (Kalra et al., 2025) apply different norms for different layers to achieve better performance. Below we systematically analyze the effect of each normalization scheme separately.

**Computational cost of different normalizations.** In terms of the computational cost, the normalization techniques discussed in equation 6 can be quite different. In particular, singular-value normalization requires computing full SVD, which is the most time consuming compared to the rest. Even though efficient approximation methods of SVD has been studied in Jordan et al. (2024); Ma et al. (2024) (e.g., the Newton-Schulz (NS) procedure), they are still much more time consuming as compare with the other three normalization techniques, see Table 1 for a test on the time required for different normalizations.

**Experimental comparison of SGD with different gradient normalizations.** We conduct the preliminary experiment of pretraining LLaMA models (See Section 4 for the experiment setting) using SGD with different normalizations applied to all the layers as specified in equation 6. We report the pretraining perplexities in Table 2, and we notice that all the normalizations improve over SGD, however none of them alone could match the performance of Stable-SPAM (which is a stabilized version of Adam). In particular, the four normalization methods can be categorized into two groups based on their performance: singular-value and column-wise normalization achieve better results than row-wise and sign normalization. We thus proceed with the group of normalizations with better performance, namely the singular-value and column-wise normalizations.

| dimension $d$ | 1024 | 2048 | 4096 |
|---|---|---|---|
| singular-value | 79.77 | 354.27 | 1958.66 |
| singular-value (NS) | 6.03 | 7.00 | 14.41 |
| column-wise | 0.10 | 0.12 | 0.17 |
| row-wise | 0.09 | 0.11 | 0.13 |
| sign | 0.03 | 0.03 | 0.03 |

Table 1: Time (ms) consumed by each of the normalization methods on a torch matrix tensor with dimension $d_{\text{in}} = d_{\text{out}} = d$, testing on a single NVIDIA A40 GPU. Here the singular-value normalization are computed both exactly (first row, using `torch.linalg.svd` directly) and inexactly using Newton-Schulz (NS) iteration (second row, see Jordan et al. (2024) for details)[3].

| | 60M | 130M | 350M |
|---|---|---|---|
| Tokens | 1.4B | 2.6B | 7.8B |
| Adam | 30.05 | 23.13 | 18.77 |
| Adam (Stable-SPAM) | 28.77 | 22.20 | 16.80 |
| singular-value (NS) | 34.15 | 25.25 | 18.73 |
| column-wise | 39.89 | 28.85 | 20.38 |
| row-wise | 79.27 | 37.67 | 21.63 |
| sign | 54.36 | 40.42 | 27.95 |

Table 2: Results (perplexity) of pretraining LLaMA models on C4 dataset, using different gradient normalizations, as specified in equation 6. For the singular-value normalization, we use the inexact Newton-Schulz (NS) iteration (again see Jordan et al. (2024) for details) for fast approximation.

**Why row-wise normalization performs poorly.** Table 2 shows that row-wise gradient normalization consistently achieves worse performance than column-wise. We identified that this is largely attributed to the effect of row-wise normalization to the last layer's (LM-head) gradients. Figure 3 shows that after row-wise is applied to the LM-head, some gradients obtain very high absolute values (up-to 150 in this case), which destabilizes the training. In contrast, column-wise results into a gradient distribution without extreme values, aiding stable training. We provide further discussion about the importance of column-wise for the last layer in Appendix A.10.

### 2.2 MOMENTUM IN THE LAST LAYER

First-order momentum ($m^t$ in equation 3) has been shown to effectively reduce the variance toward the true gradients; see Liu et al. (2020); Cutkosky & Mehta (2020). Although second-order

---

[1]Here we use the matrix induced norm. For any matrix $A \in \mathbb{R}^{n\times m}$, define

$$\|A\|_{a\to b} = \max_{x\in\mathbb{R}^m} \frac{\|Ax\|_b}{\|x\|_a}$$

where $\|\cdot\|_a$ and $\|\cdot\|_b$ are vector $\ell_a$ and $\ell_b$ norms.

[3]The difference of column- and row-wise normalization may originate from the way PyTorch stores tensors with strides, see this link.

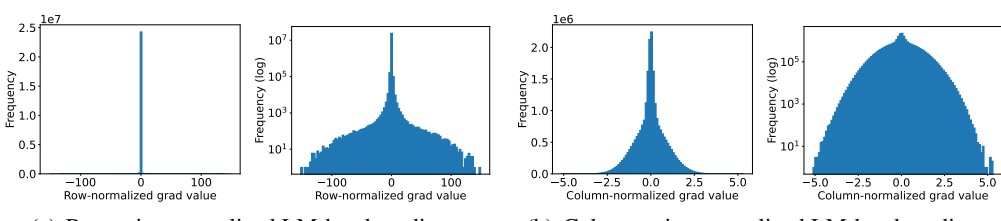

(a) Row-wise normalized LM-head gradients     (b) Column-wise normalized LM-head gradients

Figure 3: We present the histograms of the LM-head gradients after applying row-wise (a) and column-wise normalization (b). The gradients are from the 1000th training iteration of a LLaMA 130M model. It can be seen from figure (a) that row-wise results into some very high gradient values (up-to range 150 in absolute value) that we find to destabilize training.

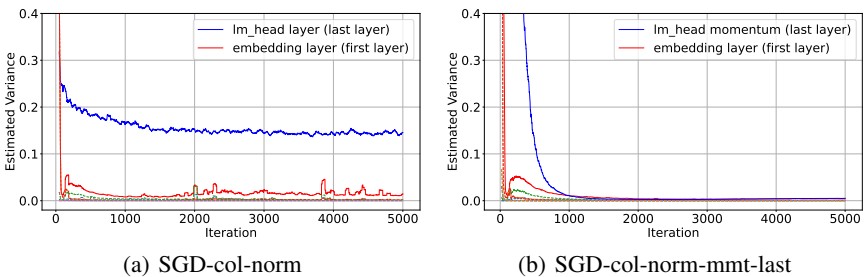

(a) SGD-col-norm        (b) SGD-col-norm-mmt-last

Figure 4: Estimated variance of the stochastic gradients (and momentum when applicable) for different layers in two methods (smoothed by 50 iterations window). We observe that when running SGD with column-wise normalization (SGD-col-norm, left plot), the variance of the last layer (lm_head) is largest for most of the time, following by the variance of the first layer (embedding) and other layers. After applying momentum to the last layer (SGD-col-norm-mmt-last, right plot), the variance of the momentum of last layer (lm_head momentum) decreases to a very low level. Interestingly, the variance of the first layer in plot (b) is also smaller than the one in plot (a).

momentum is used in adaptive optimizers such as Adam and RMSprop, more recent optimizers such as Muon have demonstrated remarkable success without second-order momentum. It is worth noticing that momentum is the major factor that introduces memory overhead in optimizer states, therefore we only consider first-order momentum in our optimizer design. Based on a minimalist optimizer design principle, we plan to investigate whether first-order momentum can be *selectively* used among different layers, while still leading to performance improvements without significantly increasing memory overhead.

**Layer-wise gradient variance**. Existing works already show the effectiveness of momentum in reducing variances (Liu et al., 2020; Cutkosky & Mehta, 2020). Naturally, if the gradients for different layers have different variances, we could achieve both memory efficiency and variance reduction by applying momentum to the layers with large variances. In the next paragraphs, we identify the layer-wise gradient variances and determine the most important layer that provides the largest performance gain when incorporating momentum; we also justify the effectiveness of different momentum parameters for different layers using a convergence analysis theory in Theorem 2.1.

We first perform a simple experiment on LLaMA 130M to check the gradient variance of different layers. To estimate the gradient variance, one needs to obtain the full gradient (by using the entire training dataset) which is not feasible in practice. Instead, we take a much larger training data batch as input[4] to estimate the true gradient. The experiment results of running SGD with column-wise normalization (SGD-col-norm) and SGD-col-norm with last layer momentum are shown in Figure 4. Interestingly, we observe that the variance of the last layer is the largest among all the layers during the training time, necessitating a specific treatment for the last layer to reduce the variance.

Next, we show theoretically that momentum helps the most for the layers with larger gradient variances. We inspect the theoretical property of applying SGD with momentum (SGD-M) to the LLM optimization problem equation 1. Note that here we did not consider column-wise normalization for

---

[4]For example, we take the training batch size as 32 and the large batch size as 512.

ease of theoretical analysis. Consider the following SGD-M algorithm to solve equation 1 ($m_l^0 = 0$ and $\theta^1$ is randomly initialized):

$$m_l^t = \beta_l m_l^{t-1} + (1 - \beta_l)g^t, \ g^t = \nabla_{\theta_l}\ell(\theta^t; \xi_t)$$
$$\theta_l^{t+1} = \theta_l^t - \eta_l m_l^t \tag{7}$$

where $t = 1, 2, ..., T$ is the iteration counter; $l = 1, ..., L$ represents different layers, i.e. we assume different layers contain different momentum with different hyperparameters. We have the following theoretical result (see Appendix A.11 for the proof).

**Theorem 2.1.** *Suppose $\ell(\theta)$ in equation 1 is lower bounded by $\ell^*$, $\gamma$-smooth (i.e. $\nabla\ell(\theta)$ is Lipschitz continuous with constant $\gamma$), also the stochastic gradient is unbiased $\mathbb{E}_{\xi_t}\nabla_{\theta_l}\ell(\theta^t; \xi_t) = \nabla_{\theta_l}\ell(\theta^t)$ and with bounded variance: $\mathbb{E}_{\xi_t}\|\nabla_{\theta_l}\ell(\theta^t; \xi_t) - \nabla_{\theta_l}\ell(\theta^t)\|^2 \leq \sigma_l^2$ for all $l = 1, ..., L$ and $t = 0, ..., T - 1$. With appropriate choice of hyperparameters $\eta_l \geq \eta$ (see equation 29 and equation 32) and $\beta_l \leq 1 - \delta$ ($\delta$ is an absolute constant), we have the following convergence result for update equation 7:*

$$\frac{1}{T}\sum_{t=1}^{T}\sum_{l=1}^{L}\mathbb{E}\|\nabla_l^t\|^2 \leq \frac{2L\gamma^{3/2}\mathbb{E}\Delta_1}{\delta^2\sqrt{T}} + \sum_{l=1}^{L}\left(\frac{1-\beta_l}{1+\beta_l}\frac{L\sqrt{\gamma}}{4\sqrt{T}} + \frac{L\gamma^{3/2}}{2\sqrt{T}} + \frac{1-\beta_l}{\beta_l^3}\frac{\gamma^2}{4LT}\right)\frac{\sigma_l^2}{\delta^2} \tag{8}$$

*where $\nabla_l^t = \nabla_{\theta_l}\ell(\theta^t; \xi_t)$ and $\Delta_1 = \ell(\theta^1) - \ell^*$.*

**Remark 2.1.** *The left-hand side of equation 8 is the expectation of the stochastic gradient, measuring the stationarity. The right-hand side suggests an overall $\mathcal{O}(1/\sqrt{T})$ rate of convergence in terms of the iteration number $T$, meanwhile the summation indicates the error to each of the layer is aggregated, whose magnitude depends on the variance level $\sigma_l$ of the specific layer.*

*Theorem 2.1 suggests that by taking different $\beta_l$ for different layers $l = 1, ..., L$, the convergence could be improved. The second term of the right-hand side of equation 8 takes the form*

$$f(x) = \frac{1-x}{1+x} + c\frac{1-x}{x^3}, \ c := \frac{\gamma}{L^2 T^{3/2}}$$

*where $x$ represents the layer-wise momentum hyperparameter $\beta_l$. It is straightforward to verify that $f(x)$ is decreasing in $(0, 1 - \delta]$, therefore an optimal strategy is to pick $\beta_l = 1 - \delta$. However from a memory-efficient point of view, only $\beta_l = 0$ saves the memory of momentums.*

*Now, if $\sigma_l$ (variance of the gradient of the $l$-th layer) is significantly higher than other layers, then taking $\beta_l$ higher than other layers will result in better convergence, which is consistent with our experimental findings. On the other hand, if $\sigma_l$ is close to) zero, taking $\beta_l = 0$ will provide memory-efficiency without harming the convergence too much. In particular, if the variances $\sigma_l \approx 0$ for $l = 1, ..., L - 1$, one could take $\beta_l = 0$ for $l = 1, ..., L - 1$ and only keep the momentum for the last layer without significantly damaging the convergence rate.*

**Proof Sketch for Theorem 2.1.** The proof follows Liu et al. (2020), with the major difference being that Liu et al. (2020) proved the case with one single layer ($L = 1$) and we extend it to multi-layer setting ($L > 1$) where each layer has difference variance level.

First, define the auxiliary sequence by

$$z_l^t := \begin{cases} \theta_l^t & t = 1 \\ \frac{1}{1-\beta_l}\theta_l^t - \frac{\beta_l}{1-\beta_l}\theta_l^{t-1} & t \geq 2 \end{cases} \tag{9}$$

we can verify two important properties: $z_l^{t+1} - z_l^t = -\eta_l g^t$ and $z_l^t - \theta_l^t = -\frac{\beta_l}{1-\beta_l}\eta_l m_l^{t-1}$ (Lemma A.3). Using $\gamma$-Lipschitz smooth, we can expand the function value difference via

$$\mathbb{E}_{\xi_t}\left[\ell\left(z^{t+1}\right)\right] \leq \ell\left(z^t\right) + \underbrace{\mathbb{E}_{\xi_t}\left[\langle\nabla\ell\left(z^t\right), z^{t+1} - z^t\rangle\right]}_{(a)} + \frac{\gamma}{2}\underbrace{\mathbb{E}_{\xi_t}\left[\|z^{t+1} - z^t\|^2\right]}_{(b)}.$$

Expanding the term (a) on the right-hand side using the update using the above properties of $z_l^t$, we will arrive at a $-\mathcal{O}(\sum_{t=1}^{T}\eta_t\sum_{l=1}^{L}\mathbb{E}\|\nabla_l^t\|^2)$ term (which is the left-hand side of equation 8), and some addition term. The additional term, combining with (b), will be bounded via Lemma A.1 and A.2, and together form the right hand side of equation 8.

---

**Algorithm 1** SCALE: Stochastic Column-normalized Last-layer Momentum

---

**Input:** Initialized trainable parameters $\theta^0$, hyperparameters $\beta_t$ and $\eta_{t,l}$.

**for** $t = 0, 1, ..., T - 1$ **do**

    Sample mini-batch data $\{\xi_{t,b}\}_{b=1,...,B}$;

    **for** Layers $l = 1, ..., L$ **do**

        Compute the stochastic gradient $g_l^t := \frac{1}{B} \sum_{b=1}^B \nabla_{\theta_l} \ell(\theta^t; \xi_{t,b})$;

        **if** $l = L$ (last layer) **then**

            $m_l^t = \beta_t m_l^{t-1} + (1 - \beta_t) g_l^t$;

        **else**

            $m_l^t = g_l^t$ (record the gradient directly);

        **end if**

        $\theta_l^{t+1} = \theta_l^t - \eta_l \mathcal{C}(m_l^t)$ where $\mathcal{C}$ is the column-wise normalization;

    **end for**

**end for**

**Output:** Trained model $\theta^T$.

---

**Experiments on different gradient normalization with last-layer-momentum**. It is observed in Table 2 that singular-value and column-wise normalization obtains better performance than the other normalizations for pre-training. We conduct another experiment to check the performance of the two types of normalization with last-layer-momentum. The results are summarized in Table 3, where we observe that as the model size gets larger, both singular-value and column-wise normalizations + last-layer-momentum are matching Adam's performance, and we choose column-wise due to the computational time concern in Table 1.

| Model Size | 60M | 130M | 350M |
|---|---|---|---|
| Tokens | 1.4B | 2.6B | 7.8B |
| Adam | 30.05 | 23.13 | 18.77 |
| Adam (Stable-SPAM) | 28.77 | 22.20 | 16.80 |
| Singular-val (NS) + mmt-last | 31.20 | 22.33 | 16.67 |
| Column-wise + mmt-last (ours) | 30.81 | 22.57 | 16.32 |

Table 3: Evaluation perplexity of two normalizations (singular-value and column-wise) when combined with last-layer-momentum (mmt-last).

## 3 THE SCALE ALGORITHM

The above investigation leads to the following simple optimizer, SCALE, detailed in Algorithm 1. The design of SCALE is motivated by empirical insights from our preceding experiments, which highlighted the importance of stabilizing updates in the last layer and controlling gradient scale via column-wise normalization. Accordingly, SCALE integrates two components: column-wise normalization of gradients and momentum updates restricted to the last layer. Despite its simplicity, SCALE is highly effective and requires only minimal modifications to standard implementations of Adam—typically just a few additional lines of code. As we will demonstrate, this lightweight design allows SCALE to outperform state-of-the-art baselines such as Adam and achieve performance competitive with Stable-SPAM and Muon, while using substantially less memory.

**Connection to existing works**. We now give a discussion on the connection of the proposed method to existing works. The proposed algorithm utilizes column-wise normalization, which is also discussed in Pethick et al. (2025). Pethick et al. (2025) uses momentum for all the layers and applies column-wise normalization as an alternative to singular-value normalization only for the last layer. Our method also differs from SWAN (Ma et al., 2024) in the following aspects: first, SWAN utilizes both the row-wise and singular-value normalization while our method only utilizes column-wise normalization, indicating certain redundancy in existing works that apply multiple normalizations; second, SWAN applies Adam for the embedding and LM head layers (the first and last layers) which increases the memory overhead, whereas our approach only introduces first-order momentum for the last layer. We summarize the techniques used in different papers in Table 4.

**Convergence of Algorithm 1**. We remark here that the convergence provided in Pethick et al. (2025, Section 5) could be adopted to provide a convergence analysis for Algorithm 1, since Algorithm 1 differs from the general framework of Pethick et al. (2025, Algorithm 2) only by the fact that Algorithm 1 applies different momentum for different layers. However, directly applying such kinds of analysis will not be able to explain why using either normalization or momentum is advantageous as compared to vanilla SGD, we choose not to present such a proof.

| Methods | Sign | Col-wise | Row-wise | Singular-val | 1st order EMA | 2nd order EMA | Memory (7B) |
|---|---|---|---|---|---|---|---|
| SGD | | | | | | | 13.48 |
| Adam | ✓ | | | | ✓ | ✓ | 40.43 |
| Muon | | | ✓ | | ✓ | | 26.95 |
| SWAN* | | | ✓ | ✓ | First/Last Layer | First/Last Layer | 14.52 |
| APOLLO* | | | | | Rank-256 | Rank-256 | 16.14 |
| APOLLO-Mini* | | | | | Rank-1 | Rank-1 | 14.53 |
| SCALE | | ✓ | | | Last Layer | | 13.74 |

*: SWAN, APOLLO and APOLLO-Mini apply Adam for the first and last layer.

Table 4: A summary of building components of related methods. "Sign", "Col-wise", "Row-wise" and "Singular-val" correspond to four normalizations in equation 6, respectively. "1st order EMA" and "2nd order EMA" stand for first and second order EMA/momentum. The last column records the memory (GB) of weights and optimizer states required for applying the corresponding methods for LLaMA 7B training (see Appendix A.2 for the details of the estimation).

## 4 EXPERIMENTS

In this section, we test the SCALE algorithm for LLM pretraining. We test on pretraining LLaMA (60M, 130M, 350M, 1B, 7B) models on the C4 (Colossal Clean Crawled Corpus) dataset (Raffel et al., 2020) and report evaluation perplexity as our metric. For all ours experiments, we follow the hyperparameter settings of Zhao et al. (2024), see Appendix A.3 for more details.

**Baselines**. We compare with Adam and its stabilized version, noted as Adam (Stable-SPAM), which performs momentum resets and clips spiked gradients (Huang et al., 2025). Among memory efficient optimizers, we compare with Muon, GaLore, Fira, APOLLO(-Mini) and SWAN (see Section 1.1 for a more detailed introduction of these works). GaLore, Fira and APOLLO(-Mini) compress the Adam states by projecting the gradients to low rank. SWAN is an optimizer that uses both singular-value and row-wise normalization, except for the first and last layers[5].

It is worth noticing that GaLore, Fira, APOLLO(-Mini) and SWAN run Adam for the first and last layers for stable training. For the 60M model the first and last layers contain over 50% of the total network parameters, and around 40% for 130M model. For the 350M this goes down to about 20% and for the 1B to about 10%. Therefore, for smaller models these methods have limited memory savings as compared to Adam, which is used to optimize a significant percentage of the network parameters.

**Results for 60M–1B models**. We report the results of the evaluation perplexity in Table 5. We can see that, despite its simplicity, SCALE outperforms existing memory efficient optimizers, also (nearly) matches the performance of Adam (Stable-SPAM) and Muon, especially for larger models, with only 35–65% of the memory. We also contrast the performance with the memory consumed in Figure 1, where we can see that the proposed method is indeed at the Pareto frontier for optimal memory usage while maintaining the (near) state-of-the-art performance. This makes it a strong candidate for large-scale pretraining under memory constraints[6].

| Steps | | 40K | 80K | 120K | 150K (final) |
|---|---|---|---|---|---|
| Tokens | | 5.2B | 10.5B | 15.7B | 19.7B |
| APOLLO† | (16.14G) | 17.55 | 14.39 | 13.23 | 13.02 |
| APOLLO-Mini† | (14.53G) | 18.03 | 14.60 | 13.32 | 13.09 |
| Muon | (26.95G) | 16.43 | 13.95 | 12.85 | 12.72 |
| SCALE (ours) | (**13.74G**) | 17.99 | 14.57 | 12.86 | **12.59** |

Table 6: Results for pretraining 7B LLaMA model on C4 dataset. †From Zhu et al. (2025).

**Results for 7B LLaMA model**. Due to limited computational resources, we run a single experiment for SCALE and Muon on 8×NVIDIA H200 141G GPUs. To compare with the reported results from APOLLO(-Mini) (Zhu et al., 2025), we train for a total of 19.7B tokens, corresponding to 150K steps. We report the final evaluation perplexity in Table 6 as well as perplexity at intermediate training steps. From the table, we conclude that SCALE outperforms Muon and APOLLO(-Mini) in terms of both final evaluation perplexity and memory.

---

[5]We copy SWAN's result from Ma et al. (2024) since we cannot replicate them due to code unavailability.

[6]Notice that we construct our optimizer step by step via a minimalist design, and we already conduct the ablations throughout our journey (see Table 2 and 3, also Figure 4). We do not conduct further ablation studies in this section, but we include experiments on throughput analysis, additional architectures, overtraining, finetuning and learning rate sensitivity in the Appendix sections A.4, A.5, A.6, A.7 and A.8 , respectively.

| Model Size | 60M | 130M | 350M | 1B |
|---|---|---|---|---|
| Tokens | 1.4B | 2.6B | 7.8B | 20B |
| Adam[†] | 30.05 (0.35G) | 23.13 (0.81G) | 18.77 (2.21G) | 15.79 (8.04G) |
| Adam (Stable-SPAM)[†] | 28.77 (0.35G) | 22.20 (0.81G) | 16.80 (2.21G) | 13.30 (8.04G) |
| Muon | **28.86** (0.23G) | **22.20** (0.54G) | 16.70 (1.47G) | 13.67 (5.36G) |
| GaLore[†] | 34.58 (0.28G) | 25.31 (0.61G) | 19.37 (1.59G) | 15.05 (4.76G) |
| Fira[†] | 30.34 (0.28G) | 22.96 (0.61G) | 16.82 (1.59G) | 14.36 (4.76G) |
| SWAN[*] | 30.00 (0.25G) | 22.83 (0.46G) | 17.14 (1.00G) | - |
| APOLLO | 30.94 (0.28G) | 22.93 (0.61G) | 16.75 (1.59G) | 14.28 (4.76G) |
| APOLLO-Mini | 31.85 (0.25G) | 23.63 (0.46G) | 17.11 (1.00G) | **13.48** (3.20G) |
| SCALE (ours) | 30.81 (**0.15G**) | 22.57 (**0.32G**) | **16.32** (**0.80G**) | **13.49** (**2.81G**) |

Table 5: Experiment results for pretraining LLaMA models on C4 dataset. The result marked [*] is from Ma et al. (2024). The results marked [†] for model sizes 60M-350M are from Glentis et al. (2025). All models are trained up-to the Chinchilla optimal number of tokens (Hoffmann et al., 2022). For Fira and APOLLO 1B runs we encountered training divergence with their default learning rates, results reported are with reduced learning rate. Among the memory-efficient optimizers we highlight the best-performing for each model size in terms of perplexity and memory. Note that SWAN does not provide a 1B model result with Chinchilla optimal tokens, but only for 13B tokens. In that setting we also achieve a superior perplexity of 14.25 (2.81G memory), compared to 14.42 (3.20G memory) of SWAN.

## 5 CONCLUSION

In this paper, we design a memory-efficient optimizer using a minimalist approach. The proposed algorithm utilizes the building blocks that lead to the success of Adam but further refines them to make it more memory efficient. We motivate each of our construction steps by theoretical or experimental evidences. The resulting algorithm, SCALE, achieves superior performance to existing memory-efficient optimizers for LLM pretraining and matches Adam while only requiring 35-45% of memory. This makes the proposed algorithm a strong candidate for large-scale pretraining under memory constraints, as well as a minimalist baseline for benchmarking more sophisticated optimizers.

## REPRODUCIBILITY STATEMENT

To ensure reproducibility, we include our code in the supplementary material. In addition, our experimental details are explained in Appendix A.3. We only use open source models and train on a publicly available dataset.

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

# A APPENDIX

## A.1 LLM ACKNOWLEDGMENT

We acknowledge the use of LLMs for grammar checking and sentence polishing.

## A.2 DETAILS OF MEMORY ESTIMATION FOR 1B AND 7B MODELS

Here we compute the memory estimate for both 1B and 7B LLaMA models. We only compute the major parameters, including embedding layers, attention and MLP layers. We follow prior works (Zhao et al., 2024; Han et al., 2024) in estimating the memory using `bfloat16` format, where each floating point number occupies 2 bytes.

**7B model**: Pre-last layers include 6.607B parameters and last layer includes 0.131B parameters, which in total leads to 6.738B parameters.

- **SGD**: Only the parameter states are stored, which amount to 13.476G memory.
- **Adam**: Apart from the parameter states, Adam/AdamW store first and second order momentum, which costs 26.952G. In total, Adam/AdamW requires 40.428G memory.
- **APOLLO**: Apart from the parameter states, APOLLO stores first-order and second-order momentum in the low-rank subspace of 256, which in total costs 16.144G.
- **APOLLO-Mini**: Apart from the parameter states, APOLLO-Mini sets rank to be 1, which leads to a total memory of 14.531G.
- **Muon**: Apart from the parameter states, Muon stores first-order momentum, which costs 13.476G. In total, Muon requires 26.952G memory.
- **SWAN**: Apart from the parameter states, SWAN additionally stores first-order and second-order momentum of the first and last layer, which costs 1.048G. In total, SWAN requires 14.524G.
- **SCALE** (Our method): Apart from parameter states, SCALE additionally stores first-order momentum of last-layer weight, which costs 0.262G. In total, SCALE requires 13.738G memory.

**1B model**: Pre-last layers include 1.273B parameters and last layer includes 0.066B parameters, which in total leads to 1.339B parameters.

- **SGD**: Only the parameter states are stored, which amount to 2.678G memory.
- **Adam**: Apart from the parameter states, Adam/AdamW store first and second order momentum, which costs 5.356G. In total, Adam/AdamW requires 8.034G memory.
- **Muon**: Apart from the parameter states, Muon stores first-order momentum, which costs 2.678G. In total, Muon requires 5.356G memory.
- **SWAN**: Apart from the parameter states, SWAN additionally stores first-order and second-order momentum of the first and last layer, which costs 0.524G. In total, SWAN requires 3.202G.
- **SCALE** (Our method): Apart from parameter states, SCALE additionally stores first-order momentum of last-layer weight, which costs 0.131G. In total, SCALE requires 2.809G memory.

## A.3 DETAILS OF THE EXPERIMENTS

For all LLaMA experiments, we follow Zhao et al. (2024) and set the sequence length to 256 and the batch size to 512, train using BF16 format and report evaluation perplexity as our metric. We also use a cosine learning rate with linear warm-up for the first 10% of the iterations. For low-rank optimizers (GaLore, Fira and APOLLO) we follow their suggested hyperparameters (including the rank) but tune the learning rates. For Muon we follow the implementation from Liu et al. (2025).

We performed wandb sweeps for all methods we tested up to models of size 350M, searching learning rates within {0.00005, 0.0001, 0.0003, 0.0005, 0.001, 0.003, 0.005, 0.01}. For the 1B model, due to resource constraints we manually tune the learning rates using as starting point the optimal learning rate from the 350M sweep. For SCALE, we use learning rate 1e-3 for models sizes 60M, 130M and 350M, 2e-4 for the 1B and 1e-4 for the 7B model. In addition, our reported result for the

1B model uses the same learning rate scaling technique used by Muon (Liu et al., 2025). Also, we set the last layer's momentum parameter $\beta = 0.9$, being a common choice for first order momentum. In addition, for all vector parameters we employ the Adam optimizer, following Jordan et al. (2024); Liao et al. (2024). This does not influence the memory usage because the vector parameters are orders of magnitude smaller in size compared to the matrix parameters.

## A.4 Training Throughput Comparison of Different Methods

We conduct a throughput (tokens/sec) comparison of the different optimizers for training LLaMA 1B on 4xH100 GPUs using the same settings as specified in A.3. Table 7 shows that our method is about 18.5% faster than singular-value (NS) based normalization methods (Muon/SWAN) and about 8% faster than GaLore/Fira which project the Adam states to low-rank using SVD. Moreover, it achieves similar throughput to Adam/Stable-SPAM and APOLLO(-Mini).

| Method | Throughput (tokens/sec) |
|---|---|
| Adam | 45019 |
| Adam (Stable-SPAM) | 44960 |
| Singular-value-based (Muon/SWAN$^\star$) | 37748 |
| GaLore | 41267 |
| Fira | 41285 |
| APOLLO | 44193 |
| APOLLO-Mini | 44567 |
| SCALE | 44728 |

Table 7: Throughput comparison of different methods for training LLaMA 1B on 4xH100 GPUs. $^\star$ : Due to code unavailability we cannot directly test the throughput of SWAN, instead we report the throughput using the Newton-Schulz (NS) approximation of Singular-value normalization (see Jordan et al. (2024) for details).

## A.5 Pretraining Results on Additional LLM Architectures

For our main experiments we choose the LLaMA family of models as it is commonly used to benchmark memory efficient optimizers for LLM pretraining (such as in GaLore (Zhao et al., 2024), Fira (Chen et al., 2024), SWAN (Ma et al., 2024) and APOLLO (Zhu et al., 2025)), which is the scope of our paper. However, to provide further evidence of the generality of our algorithm among LLM architectures, we conducted additional pretraining experiments with Qwen2-500M (Yang et al., 2024), GPT2-Medium (355M) (Radford et al., 2019) and Gemma-2B (Team et al., 2024). We followed the same experimental settings as in our LLaMA experiments, described in A.3. We provide the results bellow:

| Model | Qwen2-500M | GPT2-M (355M) |
|---|---|---|
| Tokens | 10B | 7.8B |
| Adam | 17.61 (2.96G) | 20.73 (2.13G) |
| Adam (Stable-SPAM) | 15.91 (2.96G) | 18.90 (2.13G) |
| Muon | 16.03 (1.98G) | 19.61 (1.42G) |
| GaLore | 18.22 (1.94G) | 23.66 (1.22G) |
| Fira | 15.94 (1.94G) | 19.41 (1.22G) |
| APOLLO | 16.04 (1.94G) | 19.30 (1.22G) |
| APOLLO-Mini | 16.17 (1.53G) | 19.99 (0.92G) |
| SCALE | **15.57 (1.26G)** | **19.00 (0.81G)** |

Table 8: Experiment results from pretraining Qwen and GPT2 models on the C4 dataset.

From Table 8 we observe that our method still achieves Adam-like performance while using significantly less memory and continues to outperform SOTA memory-efficient methods. We believe that

our simple, yet systematic, optimizer design makes our method more robust and less likely to overfit to specific LLM architectures.

| Model | Gemma-2B |
|---|---|
| Tokens | 32B |
| APOLLO | 12.05 (9.09G) |
| SCALE | **11.96 (6.06G)** |

Table 9: Experiment results from pretraining Gemma-2B on the C4 dataset. Due to resource and time constrains we limit our comparison to APOLLO, being the strongest baseline.

From table 9 we can see that our method achieves a lower perplexity with an even smaller memory footprint than APOLLO, which further showcases its SOTA performance as a memory-efficient pretraining algorithm.

## A.6 RESULTS ON OVERTRAINING REGIME

| Chinchilla ratio | | $1\times$ | $2\times$ | $4\times$ |
|---|---|---|---|---|
| Adam | (2.21G) | 18.77 | 17.60 | 17.21 |
| Adam (Stable-SPAM) | (2.21G) | 16.80 | 15.85 | 15.11 |
| Muon | (1.47G) | 16.70 | 15.81 | 15.18 |
| GaLore | (1.59G) | 19.37 | 18.40 | 17.81 |
| Fira | (1.59G) | 16.82 | 15.82 | 15.31 |
| APOLLO | (1.59G) | 16.75 | 15.76 | 15.06 |
| APOLLO-Mini | (1.00G) | 17.11 | 16.02 | 15.21 |
| SCALE | **(0.80G)** | **16.32** | **15.33** | **14.77** |

Table 10: Results from training the 350M LLaMA model on C4 using different token budgets. Chinchilla ratio $1\times$ corresponds to (roughly) our default token budget for the given model, i.e., 7.8B tokens (following Zhao et al. (2024)), $2\times$ to 15.6B tokens and $4\times$ to 31.2B tokens. Our method maintains its SOTA memory-efficient pretraining performance among the different training budgets.

## A.7 FINE-TUNING RESULTS

First, we want to emphasize that our optimizer design is focused on memory-efficient LLM pretraining. As pretraining is by far the most computationally expensive part of the LLM training process (for example pretraining LLaMA2-70B took about 1.72 million A100 GPU hours (Touvron et al., 2023)), it is commonly believed that an efficient SOTA pretraining optimizer is of great importance.

Instead, for fine-tuning, existing Parameter-efficient fine-tuning (PEFT) methods already reduce the memory and compute requirements substantially while giving performance competitive to full fine-tuning. Therefore we don't expect our optimizer to replace such approaches. However, to give further evidence of its generality, we provide some preliminary results on GLUE benchmarks with a pretrained RoBERTa-base (Liu et al., 2019) which we fine-tuned with our method and Adam (full fine-tuning), following the setup of Zhao et al. (2024). We conduct a learning rate search for both methods and report the best epoch results (higher the better).

| Method / Benchmark | | RTE | CoLA | MRPC | STS-B | SST-2 | QNLI | **Avg** |
|---|---|---|---|---|---|---|---|---|
| Adam | (0.75G) | 79.53 | 63.37 | 93.01 | 91.26 | 94.26 | 92.67 | 85.68 |
| SCALE | (0.33G) | 80.46 | 63.02 | 92.53 | 91.23 | 93.58 | 92.23 | 85.51 |

Table 11: Results from fine-tuning RoBERTa-base on different GLUE benchmarks (Wang et al., 2018). Memory consumption is indicated in the parenthesis.

We observe that our method, despite being designed for pretraining, can give results comparable to Adam for fine-tuning tasks while using over 2 times less memory (included in parenthesis). Again, we want to point out that our optimizer is aimed at memory-efficient pretraining where its advantages are most relevant. As a future work, it might be worth investigating combining SCALE with PEFT methods, although this is out of the scope of this paper.

## A.8 LEARNING RATE SENSITIVITY ANALYSIS

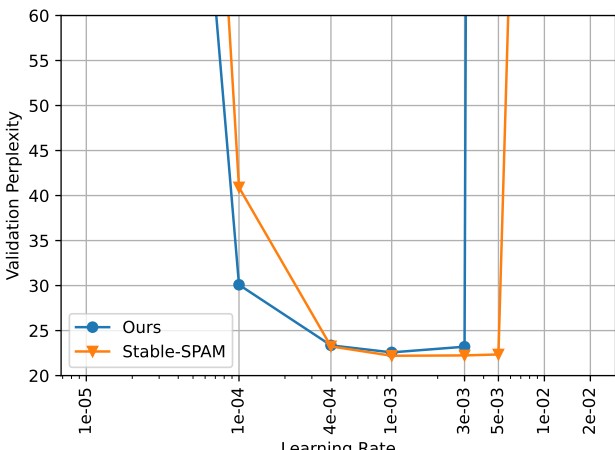

Figure 5: Learning rate sensitivity analysis, comparing Stable-SPAM (a stabilized version of Adam) and our method. Results from the 130M LLaMA model.

In Figure 5 we test the performance of our algorithm, SCALE, with different learning rates and compare it with that of Adam (Stable-SPAM version) (Huang et al., 2025). We observe that both algorithms behave similarly with a reasonable range of learning rates.

## A.9 CURVES OF TRAINING ITERATION VERSUS PERPLEXITY

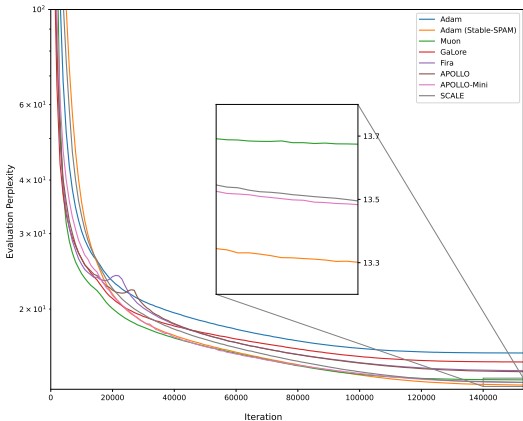

Figure 6: The evaluation perplexity curves of different methods on LLaMA 1B pretraining. Note that Muon is converging the fastest at the beginning stage, while SCALE, Adam (Stable-SPAM) and APOLLO-Mini catch up in the final stage of training.

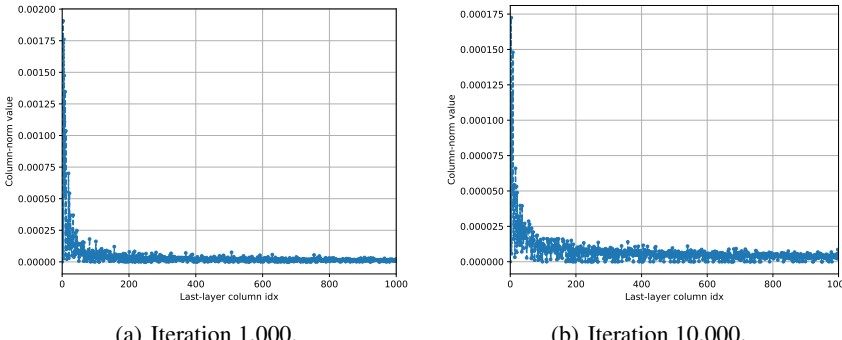

(a) Iteration 1,000.                    (b) Iteration 10,000.

Figure 7: We present the column-norm values of the last layer's gradient matrix at iterations 1,000 and 10,000 during the training of LLaMA 130M. The last layer of LLaMA 130M has gradient dimensions $d_{model} \times |V| = 768 \times 32,000$, which means that there are $|V| = 32,000$ columns (and therefore $|V|$ corresponding column-norms). For clarity we limit the x axis to the first 1000 columns. We also note that due to the design of the SentencePiece tokenizer (Kudo & Richardson, 2018) used, lower token ids (and therefore lower last-layer column-ids) generally correspond to more frequent tokens (such as "_the", "_and", "_to", etc., as we observed). Therefore, from the above figures it can be seen that more frequent tokens have much larger column-norms, potentially leading to imbalanced learning. We hypothesize that the effectiveness of column-wise is related to mitigating this phenomenon by normalizing all the column norms to the same level and thus enabling more balanced training dynamics.

### A.10 DISCUSSION ON WHY COLUMN-WISE MATTERS FOR THE LAST LAYER

In this section we further investigate the importance of column-wise normalization for the last layer, and also ablate several "mixed" normalization schemes, where more than one type of normalization is used depending on the layer.

| Method | Perplexity |
|---|---|
| 1. SCALE (all column, mmt-last) | **22.57** |
| 2. column-last, row-rest, mmt-last | 23.27 |
| 3. row-first, column-rest, mmt-last | 22.94 |
| 4. norm along larger dim, mmt-last | 23.52 |
| 5. row-last, column-rest, mmt-last | 28.83 |

Table 12: Ablation results from training the 130M LLaMA model on C4 using "mixed" normalization schemes comapred to SCALE.

All results of Table 12 are from pretraining the LLaMA 130M model using momentum only for the last layer (mmt-last, as in SCALE). Methods 1-4 all do column-wise normalization for the last layer; only method 5 (row-last, column-rest, mmt-last) does row-wise for the last layer (row-last). Note that "column-rest" or "row-rest" means we do column- or row-wise normalizations for all other layers we don't explicitly state.

We can clearly observe from the above results that row-last significantly degrades the performance. In addition, we can see that excluding the last layer, the rest of the layers are less influenced by the normalization direction. However, a uniform normalization approach as in SCALE (all column-wise) still outperforms other "mixed" normalization approaches.

To explain the above results, our intuition suggests that the last layer in LLMs is somehow "special". Indeed, it is the linear projection layer (often called Language modeling head) that maps the last hidden state (of dimension $d_{model}$) to the vocabulary logits (of dimension $|V|$ being the vocabulary size). Its gradient matrix shape is $d_{model} \times |V|$, i.e., it has $|V|$ columns, same as its weight matrix shape. Therefore, each gradient column of the last layer has a "physical meaning", i.e., it corresponds to one of the $|V|$ vocabulary tokens.

We have observed that for certain (few) tokens, their corresponding last-layer column gradients have much larger norms compared to the rest. We have also observed that such columns correspond

to more frequent tokens (see Figure 7 details). We hypothesize that such last-layer column-norm differences can potentially lead to uneven learning (i.e., more rare tokens receive smaller gradients and therefore are not "learned" by the model) or even training divergence (due to the large gradients corresponding to frequent tokens). In connection to this, Kunstner et al. (2024), aiming at explaining the large Adam-SGD gap in LLM training, observed that SGD is able to make much less progress on learning low-frequency classes compared to Adam. Based on those insights, we conclude that by normalizing the columns of the last layer our method obtains more stable training dynamics and a more "even" token learning behavior, even without using full per-parameter adaptivity as in Adam.

Although still at a preliminary stage, we believe that our above insights for the importance of column-wise normalization for the last layer can provide a deeper understanding of the normalization component and (at least to some degree) explain why SCALE is able to perform comparable to Adam for LLM pretraining.

### A.11 PROOFS FOR SECTION 2.2

In this section, we conduct the proof for Theorem 2.1. The proof follows Liu et al. (2020). First, we have the following lemmas, which are variations of Liu et al. (2020, Lemma 1 and 2).

**Lemma A.1.** *Suppose that the assumptions in Theorem 2.1 hold. For SGD-M equation 7, we have*

$$\mathbb{E}\left[\left\|m_l^t - (1-\beta_l)\sum_{i=1}^{t}\beta_l^{t-i}\nabla_{\theta_l}\ell(\theta^i)\right\|^2\right] \leq \frac{1-\beta_l}{1+\beta_l}\left(1-\beta_l^{2t}\right)\sigma_l^2. \tag{10}$$

**Proof.** It is straightforward to see $m_l^t = (1-\beta_l)\sum_{i=1}^{t}\beta_l^{t-i}g_l^i$, where $g_l^i := \nabla_{\theta_l}f(\theta^i;\xi_i)$.

We have

$$\mathbb{E}\left[\left\|m_l^t - (1-\beta_l)\sum_{i=1}^{t}\beta_l^{t-i}\nabla_{\theta_l}\ell(\theta^i)\right\|^2\right] = (1-\beta_l)^2\mathbb{E}\left\|\sum_{i=1}^{t}\beta_l^{t-i}(g_l^i - \nabla_{\theta_l}\ell(\theta^i))\right\|^2.$$

Therefore

$$\mathbb{E}\left[\left\|m_l^t - (1-\beta_l)\sum_{i=1}^{t}\beta_l^{t-i}\nabla_{\theta_l}\ell(\theta^i)\right\|^2\right]$$

$$= (1-\beta_l)^2\mathbb{E}_{\xi_1}\mathbb{E}_{\xi_2}\cdots\mathbb{E}_{\xi_t}\left\|\sum_{i=1}^{t}\beta_l^{t-i}(g_l^i - \nabla_{\theta_l}\ell(\theta^i))\right\|^2$$

$$= (1-\beta_l)^2\mathbb{E}_{\xi_1}\mathbb{E}_{\xi_2}\cdots\mathbb{E}_{\xi_t}\left[\sum_{i=1}^{t}\sum_{j=1}^{t}\langle\beta_l^{t-i}(g_l^i - \nabla_{\theta_l}\ell(\theta^i)), \beta_l^{t-j}(g_l^j - \nabla_{\theta_l}\ell(\theta^j))\rangle\right].$$

Due to unbiasedness of stochastic gradients, the cross terms cancel, therefore we have

$$\mathbb{E}\left[\left\|m_l^t - (1-\beta_l)\sum_{i=1}^{t}\beta_l^{t-i}\nabla_{\theta_l}\ell(\theta^i)\right\|^2\right]$$

$$= (1-\beta_l)^2\sum_{i=1}^{t}\beta_l^{2(t-i)}\mathbb{E}_{\xi_1}\mathbb{E}_{\xi_2}\cdots\mathbb{E}_{\xi_t}\left\|g_l^i - \nabla_{\theta_l}\ell(\theta^i)\right\|^2 \leq \frac{1-\beta_l}{1+\beta_l}(1-\beta_l^{2t})\sigma_l^2,$$

for all layers $l = 1,...,L$. $\qquad\square$

**Lemma A.2.** *Suppose that the assumptions in Theorem 2.1 hold. For SGD-M equation 7, we have*

$$\mathbb{E}\left[\left\|\frac{1-\beta_l}{1-\beta_l^t}\sum_{i=1}^{t}\beta_l^{t-i}\nabla_{\theta_l}\ell(\theta^i) - \nabla_{\theta_l}\ell(\theta^t)\right\|^2\right] \leq \sum_{i=1}^{t-1}a_{l,t,i}\mathbb{E}\left[\left\|\theta^{i+1}-\theta^i\right\|^2\right], \tag{11}$$

*for all layers $l = 1, ..., L$, where*

$$a_{l,t,i} = \frac{\gamma^2 \beta_l^{t-i}}{1 - \beta_l^t} \left( t - i + \frac{\beta_l}{1 - \beta_l} \right).$$ (12)

**Proof.** Since

$$\mathbb{E} \left[ \left\| \frac{1 - \beta_l}{1 - \beta_l^t} \sum_{i=1}^{t} \beta_l^{t-i} \nabla_{\theta_l} \ell(\theta^i) - \nabla_{\theta_l} \ell(\theta^t) \right\|^2 \right]$$

$$= \left( \frac{1 - \beta_l}{1 - \beta_l^t} \right)^2 \sum_{i,j=1}^{t} \mathbb{E} \left[ \left\langle \beta_l^{t-i} \left( \nabla_{\theta_l} \ell(\theta^t) - \nabla_{\theta_l} \ell(\theta^i) \right), \beta_l^{t-j} \left( \nabla_{\theta_l} \ell(\theta^t) - \nabla_{\theta_l} \ell(\theta^j) \right) \right\rangle \right]$$

$$\leq \left( \frac{1 - \beta_l}{1 - \beta_l^t} \right)^2 \sum_{i,j=1}^{t} \beta_l^{2t-i-j} \left( \frac{1}{2} \mathbb{E} \left[ \left\| \nabla_{\theta_l} \ell(\theta^t) - \nabla_{\theta_l} \ell(\theta^i) \right\|^2 \right] + \frac{1}{2} \mathbb{E} \left[ \left\| \nabla_{\theta_l} \ell(\theta^t) - \nabla_{\theta_l} \ell(\theta^j) \right\|^2 \right] \right)$$

$$= \left( \frac{1 - \beta_l}{1 - \beta_l^t} \right)^2 \sum_{i=1}^{t} \left( \sum_{j=1}^{t} \beta_l^{2t-i-j} \right) \frac{1}{2} \mathbb{E} \left[ \left\| \nabla_{\theta_l} \ell(\theta^t) - \nabla_{\theta_l} \ell(\theta^j) \right\|^2 \right]$$

$$+ \left( \frac{1 - \beta_l}{1 - \beta_l^t} \right)^2 \sum_{j=1}^{t} \left( \sum_{i=1}^{t} \beta_l^{2t-i-j} \right) \frac{1}{2} \mathbb{E} \left[ \left\| \nabla_{\theta_l} \ell(\theta^t) - \nabla_{\theta_l} \ell(\theta^i) \right\|^2 \right]$$

$$= \left( \frac{1 - \beta_l}{1 - \beta_l^t} \right)^2 \sum_{i=1}^{t} \frac{\beta_l^{t-i} (1 - \beta_l^t)}{1 - \beta_l} \mathbb{E} \left[ \left\| \nabla_{\theta_l} \ell(\theta^t) - \nabla_{\theta_l} \ell(\theta^i) \right\|^2 \right]$$

$$= \frac{1 - \beta_l}{1 - \beta_l^t} \sum_{i=1}^{t} \beta_l^{t-i} \mathbb{E} \left[ \left\| \nabla_{\theta_l} \ell(\theta^t) - \nabla_{\theta_l} \ell(\theta^i) \right\|^2 \right]$$

$$\leq \frac{1 - \beta_l}{1 - \beta_l^t} \sum_{i=1}^{t} \beta_l^{t-i} (t-i) \sum_{j=i}^{t} \mathbb{E} \left[ \left\| \nabla_{\theta_l} \ell(\theta^{j+1}) - \nabla_{\theta_l} \ell(\theta^j) \right\|^2 \right]$$

where we use Cauchy-Schwarz inequality for the first inequality and the last is by AM-GM inequality. Now applying the Lipschitz smoothness assumption, we have

$$\mathbb{E} \left[ \left\| \frac{1 - \beta_l}{1 - \beta_l^t} \sum_{i=1}^{t} \beta_l^{t-i} \nabla_{\theta_l} \ell(\theta^i) - \nabla_{\theta_l} \ell(\theta^t) \right\|^2 \right]$$

$$\leq \frac{1 - \beta_l}{1 - \beta_l^t} \sum_{i=1}^{t} \beta_l^{t-i} (t-i) \sum_{j=i}^{t} \gamma^2 \mathbb{E} \left[ \left\| \theta^{j+1} - \theta^j \right\|^2 \right]$$

$$\leq \frac{1 - \beta_l}{1 - \beta_l^t} \sum_{j=1}^{t-1} \left( \sum_{i=1}^{j} \beta_l^{t-i} (t-i) \right) \gamma^2 \mathbb{E} \left[ \left\| \theta^{j+1} - \theta^j \right\|^2 \right].$$

Now define

$$a'_{l,t,i} = \frac{1 - \beta_l}{1 - \beta_l^t} \gamma^2 \sum_{i=1}^{j} \beta_l^{t-i} (t-i) = \frac{\gamma^2 \beta_l^t}{1 - \beta_l^t} \left( -(t-1) - \frac{1}{1 - \beta_l} \right) + \frac{\gamma^2 \beta_l^{t-j}}{1 - \beta_l^t} \left( t - j + \frac{\beta_l}{1 - \beta_l} \right) \leq a_{l,t,i},$$

then we get equation 11. $\qquad\square$

**Lemma A.3.** *For SGD-M equation 7, define the auxiliary sequence by*

$$z_l^t := \begin{cases} \theta_l^t & t = 1 \\ \frac{1}{1 - \beta_l} \theta_l^t - \frac{\beta_l}{1 - \beta_l} \theta_l^{t-1} & t \geq 2 \end{cases}$$ (13)

*and denote $z^t = [z_1^t, ..., z_L^t]$ the entire auxiliary variable at iteration $t$. Then we have*

$$z_l^{t+1} - z_l^t = -\eta_l g_l^t$$

*and*

$$z_l^t - \theta_l^t = -\frac{\beta_l}{1 - \beta_l} \eta_l m_l^{t-1}.$$

**Proof.** For $t = 1$, we have (since $m^0 = 0$)

$$z_l^2 - z_l^1 = \frac{1}{1 - \beta_l} \theta_l^2 - \frac{\beta_l}{1 - \beta_l} \theta_l^1 - \theta_l^1 = \frac{1}{1 - \beta_l} (\theta_l^2 - \theta_l^1) = -\eta_l g_l^1.$$

For $t \geq 2$, we have

$$\begin{aligned}
z_l^{t+1} - z_l^1 &= \frac{1}{1 - \beta_l} (\theta_l^{t+1} - \theta_l^t) - \frac{\beta_l}{1 - \beta_l} (\theta_l^t - \theta_l^{t-1}) \\
&= \frac{1}{1 - \beta_l} (-\eta_l m_l^t) - \frac{\beta_l}{1 - \beta_l} (-\eta_l m_l^{t-1}) \\
&= \frac{1}{1 - \beta_l} (-\eta_l m_l^t + -\eta_l \beta_l m_l^{t-1}) \\
&= -\eta_l g_l^t.
\end{aligned}$$

For $z_l^t - \theta_l^t$, it can be computed similarly. $\qquad\square$

We need the following proposition to show the final convergence of Theorem 2.1.

**Proposition A.1.** *Suppose that the assumptions in Theorem 2.1 hold. For SGD-M equation 7, we have*

$$\mathbb{E}\left[\ell\left(z^{t+1}\right)\right]$$

$$\leq \mathbb{E}\left[\ell\left(z^t\right)\right] + \rho_0 \left(\sum_{l=1}^L \eta_l\right) \sum_{l=1}^L \left(\eta_l^2 (\frac{\beta_l}{1 - \beta_l})^2 \gamma^2 \frac{1 - \beta_l}{1 + \beta_l}\right) \sigma_l^2 + \sum_{l=1}^L \frac{\gamma \eta_l^2}{2} \sigma_l^2$$

$$+ \sum_{l=1}^L (-\eta_l + \frac{\eta_l}{2\rho_0} + \frac{\gamma \eta_l^2}{2}) \mathbb{E}\|\nabla_l \ell(\theta^t)\|^2 + 2\rho_0 \left(\sum_{l=1}^L \eta_l\right) \sum_{l=1}^L \left(\eta_l^2 (\frac{\beta_l}{1 - \beta_l})^2 \gamma^2 (1 - \beta_l^{t-1})^2 \mathbb{E}\left[\left\|\nabla_l \ell(\theta^t)\right\|^2\right]\right)$$

$$+ 2\rho_0 \left(\sum_{l=1}^L \eta_l\right) \sum_{l=1}^L \left(\eta_l^2 \gamma^2 (\frac{1 - \beta_l^t}{1 - \beta_l})^2 \mathbb{E}\left[\left\|\frac{1 - \beta_l}{1 - \beta_l^t} \sum_{i=1}^t \beta_l^{t-i} \nabla_l \ell(\theta^i) - \nabla_l \ell(\theta^t)\right\|^2\right]\right) \tag{14}$$

*where the auxiliary sequence $z_l^t$ is defined by equation 13.*

**Proof.** By Lemma A.3 we have that

$$z_l^{t+1} - z_l^t = -\eta_l g_l^t$$

and

$$z_l^t - \theta_l^t = -\frac{\beta_l}{1 - \beta_l} \eta_l m_l^{t-1},$$

for all $l = 1, 2, ..., L$.

Now using the Lipschitz smooth of $\ell$, we get

$$\begin{aligned}
\mathbb{E}_{\xi_t}\left[\ell\left(z^{t+1}\right)\right] &\leq \ell\left(z^t\right) + \mathbb{E}_{\xi_t}\left[\langle \nabla \ell\left(z^t\right), z^{t+1} - z^t \rangle\right] + \frac{L}{2} \mathbb{E}_{\xi_t}\left[\left\|z^{t+1} - z^t\right\|^2\right] \\
&= \ell\left(z^t\right) + \sum_{l=1}^L \mathbb{E}_{\xi_t}\left[\langle \nabla_{\theta_l} \ell\left(z^t\right), -\eta_l g_l^t \rangle\right] + \sum_{l=1}^L \frac{\gamma \eta_l^2}{2} \mathbb{E}_{\xi_t}\left[\left\|g_l^t\right\|^2\right] \\
&= \ell\left(z^t\right) + \sum_{l=1}^L \left[\langle \nabla_{\theta_l} \ell\left(z^t\right), -\eta_l \nabla_l \ell(\theta^t) \rangle\right] + \sum_{l=1}^L \frac{\gamma \eta_l^2}{2} \mathbb{E}_{\xi_t}\left[\left\|g_l^t\right\|^2\right]
\end{aligned}$$

where we use the unbiasedness of the gradient estimator in the last line. Now we bound the second term as follows:

$$
\begin{aligned}
&\mathbb{E}\left[\langle\nabla_{\theta_l}\ell\left(z^t\right),-\eta_l\nabla_l\ell(\theta^t)\rangle\right]\\
=&\mathbb{E}\left[\langle\nabla_{\theta_l}\ell\left(z^t\right)-\nabla_l\ell(\theta^t),-\eta_l\nabla_l\ell(\theta^t)\rangle\right]-\eta_l\mathbb{E}\|\nabla_l\ell(\theta^t)\|^2\\
\leq&\eta_l\frac{\rho_0}{2}\gamma^2\mathbb{E}\|z^t-\theta^t\|^2+(\frac{\eta_l}{2\rho_0}-\eta_l)\mathbb{E}\|\nabla_l\ell(\theta^t)\|^2\\
=&\eta_l\frac{\rho_0}{2}\sum_l\left(\eta_l^2\mathbb{E}\left[\left(\frac{\beta_l}{1-\beta_l}\right)^2\gamma^2\|m_l^{t-1}\|^2\right]\right)+(\frac{\eta_l}{2\rho_0}-\eta_l)\mathbb{E}\|\nabla_l\ell(\theta^t)\|^2
\end{aligned}
$$

where we use Cauchy-Schwarz inequality and $\rho_0$ is a positive constant to be determined.

Therefore, we get

$$
\begin{aligned}
\mathbb{E}_{\xi_t}\left[\ell\left(z^{t+1}\right)\right]\leq&\ell\left(z^t\right)+\frac{\rho_0}{2}\left(\sum_{l=1}^L\eta_l\right)\sum_{l=1}^L\left(\eta_l^2(\frac{\beta_l}{1-\beta_l})^2\gamma^2\mathbb{E}\|m_l^{t-1}\|^2\right)\\
&+\sum_{l=1}^L(\frac{\eta_l}{2\rho_0}-\eta_l)\mathbb{E}\|\nabla_l\ell(\theta^t)\|^2+\sum_{l=1}^L\frac{\gamma\eta_l^2}{2}\mathbb{E}_{\xi_t}\left[\|g_l^t\|^2\right]
\end{aligned}
\tag{15}
$$

Now from Lemma A.1 we have

$$
\begin{aligned}
\mathbb{E}\left[\|m_l^{t-1}\|^2\right]\leq&2\mathbb{E}\left[\left\|m_l^{t-1}-(1-\beta_l)\sum_{i=1}^{t-1}\beta_l^{t-1-i}\nabla_l\ell(\theta^i)\right\|^2\right]+2\mathbb{E}\left[\left\|(1-\beta_l)\sum_{i=1}^{t-1}\beta_l^{t-1-i}\nabla_l\ell(\theta^i)\right\|^2\right]\\
\leq&2\frac{1-\beta_l}{1+\beta_l}\sigma_l^2+2\mathbb{E}\left[\left\|(1-\beta_l)\sum_{i=1}^{t-1}\beta_l^{t-1-i}\nabla_l\ell(\theta^i)\right\|^2\right]
\end{aligned}
$$

and

$$
\mathbb{E}\left[\left\|\frac{1-\beta_l}{1-\beta_l^{t-1}}\sum_{i=1}^{t-1}\beta_l^{t-1-i}\nabla_l\ell(\theta^i)\right\|^2\right]\leq2\mathbb{E}\left[\|\nabla_l\ell(\theta^t)\|^2\right]+2\mathbb{E}\left[\left\|\frac{1-\beta_l}{1-\beta_l^{t-1}}\sum_{i=1}^{t-1}\beta_l^{t-1-i}\nabla_l\ell(\theta^i)-\nabla_l\ell(\theta^t)\right\|^2\right],
$$

$$
\mathbb{E}\left[\|g_l^t\|^2\right]\leq\sigma_l^2+\mathbb{E}\left[\|\nabla_l\ell(\theta^t)\|^2\right].
$$

plug these into equation 15 we get:

$$\mathbb{E}_{\xi_t}\left[\ell\left(z^{t+1}\right)\right]$$

$$\leq \ell\left(z^t\right) + \rho_0\left(\sum_{l=1}^{L}\eta_l\right)\sum_{l=1}^{L}\left(\eta_l^2(\frac{\beta_l}{1-\beta_l})^2\gamma^2\frac{1-\beta_l}{1+\beta_l}\sigma_l^2\right)$$

$$+ \rho_0\left(\sum_{l=1}^{L}\eta_l\right)\sum_{l=1}^{L}\left(\eta_l^2(\frac{\beta_l}{1-\beta_l})^2\gamma^2\mathbb{E}\left[\left\|(1-\beta_l)\sum_{i=1}^{t-1}\beta_l^{t-1-i}\nabla_l\ell(\theta^i)\right\|^2\right]\right)$$

$$+ \sum_{l=1}^{L}(\frac{\eta_l}{2\rho_0}-\eta_l)\mathbb{E}\|\nabla_l\ell(\theta^t)\|^2 + \sum_{l=1}^{L}\frac{\gamma\eta_l^2}{2}\left(\sigma_l^2 + \mathbb{E}\left[\|\nabla_l\ell(\theta^t)\|^2\right]\right)$$

$$= \ell\left(z^t\right) + \rho_0\left(\sum_{l=1}^{L}\eta_l\right)\sum_{l=1}^{L}\left(\eta_l^2(\frac{\beta_l}{1-\beta_l})^2\gamma^2\frac{1-\beta_l}{1+\beta_l}\right)\sigma_l^2 + \sum_{l=1}^{L}\frac{\gamma\eta_l^2}{2}\sigma_l^2$$

$$+ \sum_{l=1}^{L}(\frac{\eta_l}{2\rho_0}-\eta_l+\frac{\gamma\eta_l^2}{2})\mathbb{E}\|\nabla_l\ell(\theta^t)\|^2$$

$$+ \rho_0\left(\sum_{l=1}^{L}\eta_l\right)\sum_{l=1}^{L}\left(\eta_l^2(\frac{\beta_l}{1-\beta_l})^2\gamma^2(1-\beta_l^{t-1})^2\mathbb{E}\left[\left\|\frac{1-\beta_l}{1-\beta_l^{t-1}}\sum_{i=1}^{t-1}\beta_l^{t-1-i}\nabla_l\ell(\theta^i)\right\|^2\right]\right)$$

$$\leq \ell\left(z^t\right) + \rho_0\left(\sum_{l=1}^{L}\eta_l\right)\sum_{l=1}^{L}\left(\eta_l^2(\frac{\beta_l}{1-\beta_l})^2\gamma^2\frac{1-\beta_l}{1+\beta_l}\right)\sigma_l^2 + \sum_{l=1}^{L}\frac{\gamma\eta_l^2}{2}\sigma_l^2$$

$$+ \sum_{l=1}^{L}(-\eta_l+\frac{\eta_l}{2\rho_0}+\frac{\gamma\eta_l^2}{2})\mathbb{E}\|\nabla_l\ell(\theta^t)\|^2 + 2\rho_0\left(\sum_{l=1}^{L}\eta_l\right)\sum_{l=1}^{L}\left(\eta_l^2(\frac{\beta_l}{1-\beta_l})^2\gamma^2(1-\beta_l^{t-1})^2\mathbb{E}\left[\|\nabla_l\ell(\theta^t)\|^2\right]\right)$$

$$+ 2\rho_0\left(\sum_{l=1}^{L}\eta_l\right)\sum_{l=1}^{L}\left(\eta_l^2(\frac{\beta_l}{1-\beta_l})^2\gamma^2(1-\beta_l^{t-1})^2\mathbb{E}\left[\left\|\frac{1-\beta_l}{1-\beta_l^{t-1}}\sum_{i=1}^{t-1}\beta_l^{t-1-i}\nabla_l\ell(\theta^i)-\nabla_l\ell(\theta^t)\right\|^2\right]\right)$$

Now the last term above can be replaced since

$$\mathbb{E}\left[\left\|\frac{1-\beta_l}{1-\beta_l^t}\sum_{i=1}^{t}\beta_l^{t-i}\nabla_l\ell(\theta^i)-\nabla_l\ell(\theta^t)\right\|^2\right] = \mathbb{E}\left[\left\|\frac{\beta_l(1-\beta_l)}{1-\beta_l^t}\sum_{i=1}^{t-1}\beta_l^{t-1-i}\nabla_l\ell(\theta^i)-\frac{1-\beta_l^{t-1}}{1-\beta_l^t}\beta_l\nabla_l\ell(\theta^t)\right\|^2\right]$$

$$= \beta_l^2(\frac{1-\beta_l^{t-1}}{1-\beta_l^t})^2\mathbb{E}\left[\left\|\frac{1-\beta_l}{1-\beta_l^{t-1}}\sum_{i=1}^{t-1}\beta_l^{t-1-i}\nabla_l\ell(\theta^i)-\nabla_l\ell(\theta^t)\right\|^2\right]$$

$$\square$$

Now we return to the proof of Theorem 2.1.

**Proof.** [Proof of Theorem 2.1] By Proposition A.1 and Lemma A.2, we have

$$\mathbb{E}\left[\ell\left(z^{t+1}\right)\right]$$

$$\leq \mathbb{E}\left[\ell\left(z^t\right)\right] + \rho_0\left(\sum_{l=1}^{L}\eta_l\right)\sum_{l=1}^{L}\left(\eta_l^2(\frac{\beta_l}{1-\beta_l})^2\gamma^2\frac{1-\beta_l}{1+\beta_l}\right)\sigma_l^2 + \sum_{l=1}^{L}\frac{\gamma\eta_l^2}{2}\sigma_l^2$$

$$+ \sum_{l=1}^{L}(-\eta_l+\frac{\eta_l}{2\rho_0}+\frac{\gamma\eta_l^2}{2})\mathbb{E}\|\nabla_l\ell(\theta^t)\|^2 + 2\rho_0\left(\sum_{l=1}^{L}\eta_l\right)\sum_{l=1}^{L}\left(\eta_l^2(\frac{\beta_l}{1-\beta_l})^2\gamma^2(1-\beta_l^{t-1})^2\mathbb{E}\left[\|\nabla_l\ell(\theta^t)\|^2\right]\right)$$

$$+ 2\rho_0\left(\sum_{l=1}^{L}\eta_l\right)\sum_{l=1}^{L}\left(\eta_l^2\gamma^2(\frac{1-\beta_l^t}{1-\beta_l})^2\sum_{i=1}^{t-1}a_{l,t,i}\mathbb{E}\left[\|\theta^{i+1}-\theta^i\|^2\right]\right)$$

$$(16)$$

Now define the potential function:

$$\phi^t = \ell\left(z^t\right) - \ell^* + \sum_{i=1}^{t-1}\sum_{l=1}^{L} c_{l,i} \left\|\theta_l^{t+1-i} - \theta_l^{t-i}\right\|^2 \tag{17}$$

where $c_{l,i}$ are constants to be determined.

From equation 16 we get

$$\mathbb{E}\left[\phi^{t+1}\right] - \mathbb{E}\left[\phi^t\right]$$
$$\leq \rho_0 \left(\sum_{l=1}^{L}\eta_l\right)\sum_{l=1}^{L}\left(\eta_l^2(\frac{\beta_l}{1-\beta_l})^2\gamma^2\frac{1-\beta_l}{1+\beta_l}\right)\sigma_l^2 + \sum_{l=1}^{L}\frac{\gamma\eta_l^2}{2}\sigma_l^2$$
$$+ \sum_{l=1}^{L}(-\eta_l + \frac{\eta_l}{2\rho_0} + \frac{\gamma\eta_l^2}{2})\mathbb{E}\|\nabla_l\ell(\theta^t)\|^2 + 2\rho_0\left(\sum_{l=1}^{L}\eta_l\right)\sum_{l=1}^{L}\left(\eta_l^2(\frac{\beta_l}{1-\beta_l})^2\gamma^2(1-\beta_l^{t-1})^2\mathbb{E}\left[\left\|\nabla_l\ell(\theta^t)\right\|^2\right]\right)$$
$$+ 2\rho_0\left(\sum_{l=1}^{L}\eta_l\right)\sum_{l=1}^{L}\left(\eta_l^2\gamma^2(\frac{1-\beta_l^t}{1-\beta_l})^2\sum_{i=1}^{t-1}a_{l,t,i}\mathbb{E}\left[\left\|\theta^{i+1}-\theta^i\right\|^2\right]\right)$$
$$+ \sum_{l=1}^{L}c_{l,1}\mathbb{E}\left\|\theta_l^{t+1}-\theta_l^t\right\|^2 + \sum_{i=1}^{t-1}\sum_{l=1}^{L}(c_{l,i+1}-c_{l,i})\mathbb{E}\left[\left\|\theta_l^{t+1-i}-\theta_l^{t-i}\right\|^2\right] \tag{18}$$

For the term $\sum_{l=1}^{L}c_{l,1}\mathbb{E}\left\|\theta_l^{t+1}-\theta_l^t\right\|^2$, we can bound it by (denote $\nabla_l^t = \nabla_{\theta_l}\ell(\theta^t)$)

$$\sum_{l=1}^{L}c_{l,1}\mathbb{E}\left\|\theta_l^{t+1}-\theta_l^t\right\|^2 = \sum_{l=1}^{L}c_{l,1}\eta_l^2\mathbb{E}\left[\left\|m_l^t\right\|^2\right]$$
$$\leq 2\sum_{l=1}^{L}c_{l,1}\mathbb{E}\left[\left\|m_l^t - (1-\beta_l)\sum_{i=1}^{t}c_{l,1}\beta_l^{t-i}\nabla_l^i\right\|^2\right] + 2\sum_{l=1}^{L}\mathbb{E}\left[\left\|(1-\beta_l)\sum_{i=1}^{t}\beta_l^{t-i}\nabla_l^i\right\|^2\right]$$
$$\leq 2\sum_{l=1}^{L}c_{l,1}\frac{1-\beta_l}{1+\beta_l}\sigma_l^2 + 2\sum_{l=1}^{L}c_{l,1}\mathbb{E}\left[\left\|(1-\beta_l)\sum_{i=1}^{t}\beta_l^{t-i}\nabla_l^i\right\|^2\right]$$
$$\leq \sum_{l=1}^{L}c_{l,1}\eta_l^2\left(2\frac{1-\beta_l}{1+\beta_l}\sigma_l^2 + 4\mathbb{E}\left[\left\|\nabla_l^t\right\|^2\right]\left(1-\beta_l^t\right)^2\right) + 4\sum_{l=1}^{L}c_{l,1}(1-\beta_l^t)^2\eta_l^2\mathbb{E}\left[\left\|\frac{1-\beta_l}{1-\beta_l^t}\sum_{i=1}^{t}\beta_l^{t-i}\nabla_l^i - \nabla_l^t\right\|^2\right]$$
$$\leq \sum_{l=1}^{L}c_{l,1}\eta_l^2\left(2\frac{1-\beta_l}{1+\beta_l}\sigma_l^2 + 4\mathbb{E}\left[\left\|\nabla_l^t\right\|^2\right]\right) + 4\sum_{l=1}^{L}c_{l,1}(1-\beta_l^t)^2\eta_l^2\sum_{i=1}^{t-1}a_{l,t,i}\mathbb{E}\left[\left\|\theta^{i+1}-\theta^i\right\|^2\right] \tag{19}$$

where for the last three inequalities we use equation 10 and

$$\mathbb{E}\left[\left\|\frac{1-\beta_l}{1-\beta_l^t}\sum_{i=1}^{t}\beta_l^{t-i}\nabla_l^i\right\|^2\right] \leq 2\mathbb{E}\left[\left\|\nabla_l^t\right\|^2\right] + 2\mathbb{E}\left[\left\|\frac{1-\beta_l}{1-\beta_l^t}\sum_{i=1}^{t}\beta_l^{t-i}\nabla_l^i - \nabla_l^t\right\|^2\right],$$

and equation 11, respectively.

Now plugging this back to equation 18 we get:

$$\mathbb{E}\left[\phi^{t+1}\right] - \mathbb{E}\left[\phi^t\right]$$

$$\leq \rho_0 s \sum_{l=1}^{L}\left(\eta_l^2(\frac{\beta_l}{1-\beta_l})^2\gamma^2\frac{1-\beta_l}{1+\beta_l}\right)\sigma_l^2 + \sum_{l=1}^{L}\frac{\gamma\eta_l^2}{2}\sigma_l^2 + \sum_{l=1}^{L}2c_{l,1}\eta_l^2\frac{1-\beta_l}{1+\beta_l}\sigma_l^2$$

$$+ \sum_{l=1}^{L}(-\eta_l + \frac{\eta_l}{2\rho_0} + \frac{\gamma\eta_l^2}{2} + 4c_{l,1}\eta_l^2)\mathbb{E}\|\nabla_l^t\|^2 + 2\rho_0\left(\sum_{l=1}^{L}\eta_l\right)\sum_{l=1}^{L}\left(\eta_l^2(\frac{\beta_l}{1-\beta_l})^2\gamma^2(1-\beta_l^{t-1})^2\mathbb{E}\left[\|\nabla_l^t\|^2\right]\right)$$

$$+ 4\sum_{i=1}^{t-1}\sum_{l=1}^{L}c_{l,1}(1-\beta_l^t)^2\eta_l^2 a_{l,t,i}\mathbb{E}\left[\left\|\theta^{i+1}-\theta^i\right\|^2\right]$$

$$+ 2\rho_0 s\sum_{i=1}^{t-1}\sum_{l=1}^{L}\left(\eta_l^2\gamma^2(\frac{1-\beta_l^t}{1-\beta_l})^2 a_{l,t,i}\mathbb{E}\left[\left\|\theta^{i+1}-\theta^i\right\|^2\right]\right)$$

$$+ \sum_{i=1}^{t-1}\sum_{l=1}^{L}(c_{l,i+1}-c_{l,i})\mathbb{E}\left[\left\|\theta_l^{t+1-i}-\theta_l^{t-i}\right\|^2\right]$$

$$(20)$$

where we denote $s := \sum_{l=1}^{L}\eta_l$.

To make the last three lines of above non-positive, we could take

$$c_{l,i+1}\leq c_{l,i}-\left(4c_{l,1}(1-\beta_l^t)^2\eta_l^2 + 2\rho_0 s\eta_l^2\gamma^2(\frac{1-\beta_l^t}{1-\beta_l})^2\right)a_{l,t,t-i}$$

for all $l = 1, ..., L$.

We can take (since $1-\beta_l^t < 1$)

$$c_{l,i+1}=c_{l,i}-\left(4c_{l,1}\eta_l^2 + 2\rho_0 s\eta_l^2\gamma^2(\frac{1}{1-\beta_l})^2\right)\gamma^2\beta_l^i\left(i+\frac{\beta_l}{1-\beta_l}\right)$$

also we can take $c_{l,1}$ such that

$$c_{l,1}=\sum_{i=1}^{\infty}\left(4c_{l,1}\eta_l^2 + 2\rho_0 s\eta_l^2\gamma^2(\frac{1}{1-\beta_l})^2\right)\gamma^2\beta_l^i\left(i+\frac{\beta_l}{1-\beta_l}\right)$$

$$= \left(4c_{l,1}\eta_l^2 + 2\rho_0 s\eta_l^2\gamma^2(\frac{1}{1-\beta_l})^2\right)\gamma^2\left(\sum_{i=1}^{\infty}i\beta_l^i + \frac{\beta_l}{1-\beta_l}\sum_{i=1}^{\infty}\beta_l^i\right)$$

$$= \left(4c_{l,1}\eta_l^2 + 2\rho_0 s\eta_l^2\gamma^2(\frac{1}{1-\beta_l})^2\right)\frac{\gamma^2\beta_l(1+\beta_l)}{(1-\beta_l)^2}$$

i.e.

$$c_{l,1}=\frac{2\rho_0 s\eta_l^2\gamma^4\beta_l\frac{1+\beta_l}{(1-\beta_l)^4}}{1-4\eta_l^2\frac{\gamma^2\beta_l(1+\beta_l)}{(1-\beta_l)^2}}.\tag{21}$$

Note that we can give an upper bound for $c_{l,1}$ by requiring the denominator $\geq 1/2$, i.e.

$$\eta_l\leq\frac{1-\beta_l}{\gamma\sqrt{8\beta_l(1+\beta_l)}},\tag{22}$$

consequently

$$c_{l,1}\leq 4\rho_0 s\eta_l^2\gamma^4\beta_l\frac{1+\beta_l}{(1-\beta_l)^4}.\tag{23}$$

Now we have (since the last three terms of equation 20 sum to negative)

$$\mathbb{E}\left[\phi^{t+1}\right] - \mathbb{E}\left[\phi^t\right]$$

$$\leq \rho_0 s \sum_{l=1}^{L} \left( \eta_l^2 (\frac{\beta_l}{1-\beta_l})^2 \gamma^2 \frac{1-\beta_l}{1+\beta_l} \right) \sigma_l^2 + \sum_{l=1}^{L} \frac{\gamma \eta_l^2}{2} \sigma_l^2 + \sum_{l=1}^{L} 2c_{l,1} \eta_l^2 \frac{1-\beta_l}{1+\beta_l} \sigma_l^2 \tag{24}$$

$$+ \sum_{l=1}^{L} \left( -\eta_l + \frac{\eta_l}{2\rho_0} + \frac{\gamma \eta_l^2}{2} + 4c_{l,1} \eta_l^2 + 2\rho_0 s \eta_l^2 (\frac{\beta_l}{1-\beta_l})^2 \gamma^2 (1-\beta_l^{t-1})^2 \right) \mathbb{E}\|\nabla_l^t\|^2$$

Now take $\rho_0 = 2$, and take $\eta_l$ such that

$$-\frac{3}{4}\eta_l + \frac{\gamma \eta_l^2}{2} + 4c_{l,1}\eta_l^2 + 4s\eta_l^2(\frac{\beta_l}{1-\beta_l})^2\gamma^2(1-\beta_l^{t-1})^2 \leq -\frac{\eta_l}{2}, \tag{25}$$

the coefficient of $\mathbb{E}\|\nabla_l^t\|$ can be greatly simplified. Note that we can guarantee equation 25 if

$$\eta_l \leq \min\left\{ \frac{1}{8\gamma}, \frac{1}{8\gamma L}(\frac{1-\beta_l}{\beta_l})^2, \frac{1-\beta_l}{4\gamma} \sqrt[3]{\frac{1-\beta_l}{L\beta_l(1+\beta_l)}} \right\} \tag{26}$$

for all $l = 1, 2, ..., L$. Here each term in the right hand side of equation 26 is bounding each term on the left hand side of equation 25.

We get

$$\sum_{l=1}^{L} \frac{\eta_l}{2} \mathbb{E}\|\nabla_l^t\|^2 \leq \mathbb{E}\left[\phi^t\right] - \mathbb{E}\left[\phi^{t+1}\right]$$

$$+ 2s \sum_{l=1}^{L} \left( \eta_l^2(\frac{\beta_l}{1-\beta_l})^2\gamma^2 \frac{1-\beta_l}{1+\beta_l} \right) \sigma_l^2 + \sum_{l=1}^{L} \frac{\gamma \eta_l^2}{2} \sigma_l^2 + \sum_{l=1}^{L} 2c_{l,1}\eta_l^2 \frac{1-\beta_l}{1+\beta_l} \sigma_l^2. \tag{27}$$

Note that we have defined $s := \sum_{l=1}^{L} \eta_l$. We can upper bound it by $s \leq L/\sqrt{\gamma T}$ if we assume $\eta_l \leq 1/\sqrt{\gamma T}$. Combining equation 22 and equation 26 we get:

$$\eta_l \leq \min\left\{ \frac{1}{8\gamma}, \frac{1}{8\gamma L}(\frac{1-\beta_l}{\beta_l})^2, \frac{1-\beta_l}{\gamma\sqrt{8\beta_l(1+\beta_l)}}, \frac{1-\beta_l}{4\gamma} \sqrt[3]{\frac{1-\beta_l}{L\beta_l(1+\beta_l)}}, \frac{1}{\sqrt{\gamma T}} \right\}, \tag{28}$$

which is satisfied if (since $\beta_l(1+\beta_l) \leq 2$)

$$\eta_l = \min\left\{ \frac{1}{8\gamma}, \frac{1}{8\gamma L}(\frac{1-\beta_l}{\beta_l})^2, \frac{1-\beta_l}{4\gamma}, \frac{1-\beta_l}{4\gamma} \sqrt[3]{\frac{1-\beta_l}{2L}}, \frac{1}{\sqrt{\gamma T}} \right\}. \tag{29}$$

By taking the product of two of the terms in the RHS of the above relation, we have the following upper bound on $\eta_l^2$:

$$\eta_l^2 \leq \frac{1}{8\gamma L}(\frac{1-\beta_l}{\beta_l})^2 \frac{1}{\sqrt{\gamma T}}. \tag{30}$$

Plugging equation 30 and equation 23 into equation 27, we obtain:

$$\sum_{l=1}^{L} \eta_l \mathbb{E}\|\nabla_l^t\|^2 \leq 2(\mathbb{E}\left[\phi^t\right] - \mathbb{E}\left[\phi^{t+1}\right])$$

$$+ \sum_{l=1}^{L} \left( \frac{1-\beta_l}{1+\beta_l} \frac{1}{4\gamma T} + \frac{1}{2T} + \frac{1-\beta_l}{\beta_l^3} \frac{\sqrt{\gamma}}{4L^2 T^{3/2}} \right) \sigma_l^2. \tag{31}$$

Now since $\beta_l \leq 1 - \delta$, we have that $\eta_l$ is lower bounded

$$\eta_l = \min\left\{ \frac{1}{8\gamma}, \frac{1}{8\gamma L}(\frac{1-\beta_l}{\beta_l})^2, \frac{1-\beta_l}{4\gamma}, \frac{1-\beta_l}{4\gamma} \sqrt[3]{\frac{1-\beta_l}{2L}} \right\}$$

$$\geq \min\left\{ \frac{1}{8\gamma}, \frac{\delta^2}{8\gamma L}, \frac{\delta}{4\gamma}, \frac{\delta}{4\gamma} \sqrt[3]{\frac{\delta}{2L}}, \frac{1}{\sqrt{\gamma T}} \right\} =: \eta \tag{32}$$

we obtain:

$$\sum_{l=1}^{L} \mathbb{E}\|\nabla_l^t\|^2 \leq \frac{2(\mathbb{E}\left[\phi^t\right] - \mathbb{E}\left[\phi^{t+1}\right])}{\eta}$$

$$+ \frac{1}{\eta}\sum_{l=1}^{L}\left(\frac{1-\beta_l}{1+\beta_l}\frac{1}{4\gamma T} + \frac{1}{2T} + \frac{1-\beta_l}{\beta_l^3}\frac{\sqrt{\gamma}}{4L^2 T^{3/2}}\right)\sigma_l^2. \tag{33}$$

Now by telescoping sum of equation 31 for $t = 1, ..., T$, also notice that $1/\eta = \Theta(\sqrt{\gamma T}L\gamma/\delta^2)$ as $T$ grows, we get our final result of

$$\frac{1}{T}\sum_{t=1}^{T}\sum_{l=1}^{L}\mathbb{E}\|\nabla_l^t\|^2 \leq \frac{2L\gamma^{3/2}\mathbb{E}\Delta_1}{\delta^2\sqrt{T}} + \sum_{l=1}^{L}\left(\frac{1-\beta_l}{1+\beta_l}\frac{L\sqrt{\gamma}}{4\sqrt{T}} + \frac{L\gamma^{3/2}}{2\sqrt{T}} + \frac{1-\beta_l}{\beta_l^3}\frac{\gamma^2}{4LT}\right)\frac{\sigma_l^2}{\delta^2}. \tag{34}$$

This completes the proof. $\qquad\qquad\square$

