# OpenReview forum: "Memory-Efficient LLM Pretraining via Minimalist Optimizer Design"
_ICLR.cc/2026/Conference — Submitted to ICLR 2026_

### Official Review · Reviewer_CaWc · 2025-10-26

**Soundness:** 4
**Presentation:** 4
**Contribution:** 3
**Rating:** 8
**Confidence:** 3

**Summary:**

This paper tackles the problem of reducing GPU memory footprint in LLM pretraining by simplifying the optimizer. Standard adaptive optimizers like Adam require storing first and second moment statistics for each parameter which triples the memory requirement for "persistent" (i.e. non activation or gradient) tensors. The authors pose a central question: what are the minimum modifications to SGD needed to achieve SOTA pretraining performance? They pursue a bottom-up, minimalist design, systematically testing fundamental components (gradient normalization and momentum) to bridge the gap between SGD and Adam with minimal memory cost.

the paper identifies two key techniques that dramatically improve SGD:
1. Column-wise gradient normalization (normalize gradients along the output dimension of each layer), which boosts training effectiveness with a simple closed-form scaling and no extra memory use
2. First-order momentum applied only to the last layer, where gradient variance is highest, to stabilize and accelerate learning with negligible memory overhead. Combining these yields a new optimizer called SCALE, which uses roughly the same memory as vanilla SGD. Notably, SCALE achieves performance on par with Adam and other SOTA optimizers while using a fraction of the memory. For example, on a 1B-parameter LLaMA model, SCALE reaches similar final perplexity to Adam (and Muon) while using only 35–52% of the memory that Adam/Muon require. Compared to other recent memory-efficient optimizers (e.g. GaLore, Fira, Apollo), SCALE attains better perplexity with only about 59% of their memory cost on a 1B model.

In summary, the paper’s primary contributions are:
1. Defining a minimalist optimizer design approach for LLM training under memory constraints, and executing a principled study to find the smallest necessary improvements to SGD
2. Introducing SCALE, a simple two-component optimizer (column-normalized gradients + last-layer momentum) that is memory-efficient and performant
3. Empirical validation across multiple LLM scales (60M, 130M, 350M, 1B, and a partial 7B) showing SCALE matches or exceeds Adam’s performance while drastically reducing memory usage
4. Theoretical insight into why these minimal modifications suffice: the authors prove that momentum yields the most benefit on layers with high gradient variance, justifying concentrating momentum in the last layer. They also analyze different normalization schemes to explain why column-wise normalization is especially effective for stabilizing training

**Strengths:**

1. Thorough experimental approach: the authors didn’t jump straight to proposing an algorithm. Instead, they earned the design of SCALE through systematic exploration. They ran extensive ablations to dissect what matters: for example, testing four different normalization strategies across multiple model sizes (60M -> 350M) and demonstrating that while all improve upon vanilla SGD, only column-wise or singular-value normalization come close to bridging the gap. Similarly, they evaluated momentum placement by trying momentum in the last layer and showing it dramatically boosts performance, especially for larger models where _"both singular-value and column-wise normalization + last-layer momentum are matching Adam’s performance"_. This methodical approach builds confidence that the chosen techniques are indeed the critical ones. The paper effectively rules out alternatives (e.g., it shows why row-wise normalization fails by tracing it to unstable gradient distributions), which strengthens the validity of their conclusions.
2. The resulting SCALE optimizer is simple by design. It requires only minimal modifications to existing SGD/Adam implementations to achieves SOTA results. Practitioners can easily adopt SCALE without complex infrastructure changes or hyper-parameter gymnastics. The elegance of using one normalization and one localized momentum is that it’s easy to maintain and understand. The authors also connect this simplicity to existing ideas (showing how it relates to prior works but with fewer components) so the novelty doesn’t come at the cost of obscurity.
3. The empirical results demonstrate that SCALE offers outstanding memory-perplexity trade-offs at scale. The strength here is the practical efficiency gain: large savings in memory without compromise in model quality. Also the method’s benefits increase with model size. The paper notes that as model size grows, SCALE matches or even exceeds the performance of Adam and others. For example, by 350M parameters, “column-wise + last-momentum” outperforms the tuned Adam (Stable-SPAM) baseline in perplexity. At 1B, SCALE ties with the best baseline, and on a 7B partial run, SCALE achieved a lower perplexity than both Muon and Apollo-Mini under the same training length.
4. Despite introducing an extra normalization step each iteration, SCALE’s design keeps compute costs low. Appendix Table 7 shows that SCALE’s training speed in tokens/sec is essentially the same as Adam and even slightly higher in their setup. This is a significant strength because it means the memory savings do not come at the cost of slower training, which is a common pitfall for some compressed optimizers that spend extra cycles on state transformations.
5. Beyond raw performance, the work provides insights that strengthen our understanding of optimization. The identification of the last layer’s gradient variance as a key issue is backed by both an empirical plot (variance curves) and a theoretical argument (Theorem 2.1). This realistically explans why momentum should applied only to the last layer. It suggests the approach is built on solid principles (variance reduction and convergence analysis) rather than just trial-and-error.

**Weaknesses:**

While this paper is very strong, there are a few aspects that could be seen as weaknesses or areas for improvement:
1. The optimizer does not introduce fundamentally new primitives beyond what’s known. One could argue that the contribution is more in the clever combination and insight rather than a brand-new algorithmic concept as it still builds upon gradient normalization and momentum.
2. The experiments focus exclusively on transformer-based LLM pretraining (on C4 dataset). It remains unclear how well it generalizes to other domains or tasks such as vision models and post-training. This isn’t exactly a weakness of what’s done (the paper already has an impressive array of experiments), but it leaves an open question about generality.

**Questions:**

1. The study identifies the last layer as having the highest gradient variance and thus focuses momentum there. Did the authors consider or experiment with applying momentum to the first (embedding) layer as well, since Figure 4a shows the embedding layer had the next-largest variance, while at a significantly smaller scale?
2. By design, SCALE foregoes second-order moment. While the paper’s results suggest this isn’t needed for the tested scenario, are there cases where neglecting second-order adaptivity might hurt?
3. Echoing 2nd point in Weaknesses section, while the results for language model pretraining are strong, do the authors have any preliminary observations on how SCALE performs in other settings?
4. It's mentioned in appendix that SCALE in practice uses Adam (or full adaptivity) for some "vector" parts of the model that are small (and possibly critical like embeddings?). Could the authors elaborate on this choice? For example, did they notice any degradation if even those vector parameters were trained with the simplified optimizer? And are the embedding word vectors treated as “matrix” (since they are large) or “vector” in this context?
5. (Minor) the authors might consider releasing pseudocode or a snippet illustrating the few lines of change needed to implement SCALE in a standard training loop to encourage adoption

---

> ### Author Response · Authors · 2025-11-22
>
> We thank the reviewer for the very detailed feedback and the highly positive evaluation of our paper and proposed method! Bellow we provide our detailed responses to the weaknesses and questions.
>
> **Weakness 1:**
>
>
> > 1. The optimizer does not introduce fundamentally new primitives beyond what’s known. One could argue that the contribution is more in the clever combination and insight rather than a brand-new algorithmic concept as it still builds upon gradient normalization and momentum.
>
> **Response**:
>
> We agree with the reviewer that the purpose of our paper is not to propose new optimization techniques, but to systematically investigate *what are the minimum modifications to plain SGD needed to match state-of-the-art pretraining performance*. Our bottom-up approach yields novel insights about the effectiveness of column-wise normalization as well as the necessity of last-layer momentum, the latter being further supported by theoretical evidence we provide.
>
> Perhaps most striking is that this minimal set of modifications, when combined, gives rise to SCALE, an optimizer that achieves state-of-the-art memory-efficient pretraining performance and surpasses strong baselines such as Fira and APOLLO, despite its much simpler design.
>
> Therefore we believe that our study is novel and substantially advances the current understanding of what is necessary for memory-efficient pretraining optimizers.
>
> **Weakness 2:**
>
> > 2. The experiments focus exclusively on transformer-based LLM pretraining (on C4 dataset). It remains unclear how well it generalizes to other domains or tasks such as vision models and post-training. This isn’t exactly a weakness of what’s done (the paper already has an impressive array of experiments), but it leaves an open question about generality.
>
> For post-training we refer the reviewer to our answer to their 3rd question bellow. Regarding vision models, as our optimizer design is aimed at language modeling, we consider vision models ouside the scope of our study. Such vision architectures generally vary significantly from LLMs and therefore our insights and techniques may not generalize to such models. Instead, to further highlight our optimizer's generality within its intended scope (language modeling), as asked by reviewer 75sN, we pretrain additional LLM architectures and show SCALE achieves state-of-the-art results. We refer the reviewer to our answer to reviewer 75sN for details.
>
> We believe that the popularity of LLMs as well as their very resource-intensive pretraining process justifies our study's focus (memory-efficient LLM pretraining) and significance.
>
> **Question 1:**
>
> > 1. The study identifies the last layer as having the highest gradient variance and thus focuses momentum there. Did the authors consider or experiment with applying momentum to the first (embedding) layer as well, since Figure 4a shows the embedding layer had the next-largest variance, while at a significantly smaller scale?
>
> **Response**:
>
> We thank the reviewer for the question. Bellow we present some experiments where we applied momentum both to the first and the last layer (SGD column-wise mmt-(first+last)) and provide a comparison:
>
> | Method / LLaMA Model | 60M | 130M | 350M |
> |--------            |------------|--------|--------|
> | SGD column-wise (no momentum)     |  	39.89 (0.12G) | 28.85 (0.27G) | 	20.38 (0.74G) |
> | SGD column-wise mmt-last (SCALE) 	|   30.81 (0.15G) |	22.57 (0.32G) |	   16.32 (0.80G)  |
> | SGD column-wise mmt-(first+last)  |   30.35 (0.18G) |	22.26 (0.37G) |	   16.14 (0.87G)  |
>
>
> Note that the memory consumption is included in the parentheses. From the result we can see that, the additional momentum for the first layer does not result to significant gain in model performance, which further demonstrates our observation in Fig. 4 that only last-layer momentum is truly necessary. Thus, we choose only last-layer momentum following the minimalist design approach.
>
> **Question 2:**
>
> > 2. By design, SCALE foregoes second-order moment. While the paper’s results suggest this isn’t needed for the tested scenario, are there cases where neglecting second-order adaptivity might hurt?
>
> **Response**: We thank the reviewer for the question. Although the recent trend in optimizer design for LLMs suggests that we could drop seocnd-order momentum without harming the performance, e.g. Muon [1], Scion [2], existing literature suggest that second-order moment might be particularly helpful when we are facing non-stationary optimization problems such as GAN training and RL policy optimization [3].

---

> > ### Author Response · Authors · 2025-11-22
> >
> > **Question 3:**
> >
> > > 3. Echoing 2nd point in Weaknesses section, while the results for language model pretraining are strong, do the authors have any preliminary observations on how SCALE performs in other settings?
> >
> > **Response:**
> >
> > We thank the reviewer for the question. In short, we focus on LLM pretraining as it is by far the most compute-intensive type of training, thus making optimizer efficiency very important. As also asked by reviewer 75sN, we conducted some preliminary fine-tuning experiments, which we also include here for ease of reference:
> >
> > Our results are on GLUE benchmarks with a pretrained RoBERTa-base which we fine-tuned with our method and Adam (full fine-tuning), following the setup of [4]. We conduct a learning rate search for both methods and report the best epoch results (higher the better).
> >
> > | Method / Benchmark | RTE | CoLA  | MRPC | STS-B | SST-2 | QNLI | Avg |
> > |--------            |------------|--------|--------|--------|--------|--------| --------|
> > |Adam    (0.75G)      | 79.53 | 63.37 | 93.01 | 91.26 | 94.26 | 92.67 | 85.68 |
> > |SCALE   (0.33G)      | 80.46 | 63.02 | 92.53 | 91.23 | 93.58 | 92.23 | 85.51 |
> >
> > We observe that our method, despite being designed for pretraining, can give results comparable to Adam for fine-tuning tasks while using over 2 times less memory (included in parenthesis). Again, we want to point out that our optimizer is aimed at memory-efficient pretraining where its advantages are most relevant. As a future work, it might be worth investigating combining SCALE with PEFT methods, although this is out of the scope of this paper.
> >
> > As a future work, it might be worth investigating combining SCALE with PEFT methods, although this is out of the scope of this paper.
> >
> > **Question 4:**
> >
> > > 4. It's mentioned in appendix that SCALE in practice uses Adam (or full adaptivity) for some "vector" parts of the model that are small (and possibly critical like embeddings?). Could the authors elaborate on this choice? For example, did they notice any degradation if even those vector parameters were trained with the simplified optimizer? And are the embedding word vectors treated as “matrix” (since they are large) or “vector” in this context?
> >
> > **Response:**
> >
> > We thank the reviewer for the question and for carefully reading our appendix!
> >
> > We want to clarify that we meant we use Adam only for 1D layers, which consist of a single parameter vector (1D). For LLMs, such layers are usually the LayerNorm/RMSNorm  layers. Since their weights form a single vector, the gradient normalization techniques we are considering are not fully applicable. For example, singular-value based normalization cannot be applied (as it needs a 2D matrix as input) and even column or row normalizations are not well defined for a single vector (they reduce to either element-wise or layer-wise normalization, depending on whether we consider the vector a row or a column).
> >
> > Since those 1D layers do not meaningfully influence the memory consumption (for example for the 1B LLaMA model such layers only contribute to about 0.01% of the total parameters), we can still optimize them with Adam and be memory-efficient. Also note that we find such layers not to be sensitive to optimizer hyperparameters, so there is no extra tuning involved. The same practice is also followed by all other memory-efficient optimizer baselines we compare to.
> >
> > We also want to emphasize that we treated the embedding layer (i.e., the first layer, which has parameter matrix |V| x d_model where |V| is the vocabulary size and d_model is the hidden model dimension) the same as any other layer with 2D weight matrices. That is, **we applied our optimizer to the embedding layer (not Adam)**. This was important since the embedding layer has a large number of trainable parameters, so applying Adam to it would have resulted in a non-negligible increase in the memory consumption. This is in contrast to the memory-efficient optimizer baselines we compare to, which need to apply Adam for the embedding layer which increases the memory requirements.
> >
> > **Question 5:**
> >
> > > 5. (Minor) the authors might consider releasing pseudocode or a snippet illustrating the few lines of change needed to implement SCALE in a standard training loop to encourage adoption
> >
> > **Response:** We thank the reviewer for the nice suggestion, we agree that such an illustration will help showcase the simplicity of our optimizer and encourage adoption. We will include it in our final draft.
> >
> >
> > [1] Jordan, Keller, et al. "Muon: An optimizer for hidden layers in neural networks." (2024).
> >
> > [2] Pethick, Thomas, et al. "Training Deep Learning Models with Norm-Constrained LMOs." Forty-second International Conference on Machine Learning.
> >
> > [3] The Marginal Value of Adaptive Gradient Methods in Machine Learning, 2017.
> >
> > [4] GaLore: Memory-Efficient LLM Training by Gradient Low-Rank Projection, 2024.

---

### Official Review · Reviewer_agWr · 2025-10-31

**Soundness:** 3
**Presentation:** 2
**Contribution:** 3
**Rating:** 4
**Confidence:** 3

**Summary:**

This paper studies modifications of gradient descent to enable faster training for large language models. The analyze the importance of gradient normalizations and momentums, and how training is affected. Thereby, they introduce this new training scheme called SCALE, which includes column-wise gradient descent normalization, along with first-order momentum only to the final layer of the LLM. Using this method, it has been empirically shown that SCALE performs well, often better and more memory efficient than Adam optimizer on models with a large number of parameters.

**Strengths:**

- The paper is easy to understand, the results are neatly written and presented.
- There is an extensive comparison of several previously known optimization techniques along with the proposed ones.
- The algorithm for SCALE is quite simple, yet it outperforms known optimization techniques.
- The analysis of the ideas are also displayed well.

**Weaknesses:**

- From my understanding, the ideas are not very novel as most of the techniques have been used before.
- I think the proposed method is not very general. They chose to include momentum only for the last LLM layer since observed variance is high in the last layer. However, this might not always be the case, and a complete optimization scheme needs to be more adaptive to general architectures.
- It will look good to write the final term for updating \theta_t after equation 5.
- Please define the norms ||.||_{a -> b}.
- The notation is somewhat confusing, for instance, in line 383 of algorithm 1, g^t_l is computed and it's hard to tell where it's used since it doesn't appear anywhere else in the algorithm.
- The proof of theorem 2.1 is hard to follow since it contains a series of equations. It will be helpful to give a larger overview of the proof and highlighting the steps to prove them, and the purpose of the lemmas.
- Also, please mention an overview of the proofs of the lemmas, and how the inequalities follow from each other, since they are quite long and tedious to verify.

**Questions:**

- Is variance of stochastic gradient computed over batches or individual training data?
- Could you discuss the weaknesses above, especially giving proof intuitions?

---

> ### Author Response · Authors · 2025-11-22
>
> We appreciate the reviewer’s time and valuable comments. Bellow we provide our detailed response to the weaknesses and questions.
>
> **Weakness 1**
>
> > 1. From my understanding, the ideas are not very novel as most of the techniques have been used before.
>
> **Response:**
>
> The purpose of our paper is not to propose new optimization techniques, but to systematically investigate *what are the minimum modifications to plain SGD needed to match state-of-the-art pre-training performance*. Our bottom-up approach yields novel insights about the effectiveness of column-wise normalization as well as the necessity of last-layer momentum, the latter being further supported by theoretical evidence we provide.
>
> Perhaps most striking is that this minimal set of modifications, when combined, gives rise to SCALE, an optimizer that achieves state-of-the-art memory-efficient pretraining performance and surpasses strong baselines such as Fira and APOLLO, despite its much simpler design.
>
> Therefore we believe that our study is novel and substantially advances the current understanding of what is necessary for memory-efficient pretraining optimizers. We hope that the reviewer (and broader community) will recognize that the simplicity of the resulting algorithm should not diminish the originality or impact of the underlying research efforts and insights.
>
> **Weakness 2**
>
> > 2. I think the proposed method is not very general. They chose to include momentum only for the last LLM layer since observed variance is high in the last layer. However, this might not always be the case, and a complete optimization scheme needs to be more adaptive to general architectures.
>
> **Response:**
>
> In regards to the generality of our design choices, we want to emphasize that our focus is on pretraining LLM models. The vast majority of modern LLMs follows the standard design structure we study, consisting of an Embedding layer, a stuck of transformer blocks and a final linear projection layer (Language Modeling Head) that maps the hidden state to the vocabulary logits.
>
> Popular and widely used model families such as LLaMA, GPT, Qwen and Gemma (just to name a few) all are based on this particular structure and only differ in depth/width or other fine-grained design choices such as activation function types.
>
> To provide further evidence of the generality of our approach, as also asked by reviewer 75sN, we conducted new pretraining experiments using additional LLM architectures. The results validate the SOTA memory-efficient pretraining performance of our method and its generality among different LLMs, and we refer the reviewer to our response to reviewer 75sN for the details.
>
>
> **Weakness 3**
>
> > 3. It will look good to write the final term for updating \theta_t after equation 5.
>
> **Response** We thank the reviewer for the suggestion. Equation 5 is the update for Adam. So essentially we are dissecting Adam (equation 3) into equation 4 and 5. In the updated draft, we included a sentence mentioning that 4 and 5 together forms the original Adam (equation 3).
>
> **Weakness 4**
>
> > 4. Please define the norms $\|.\|_{a -> b}$.
>
> **Response** Thanks for bringing this up. Matrix induced norm is defined as: for any matrix $A\in\mathbb{R}^{n\times m}$, the matrix induced norm is defined as
>
> $$
> \left\lVert A \right\rVert_{a \to b}
> :=
> \max_{x \in \mathbb{R}^m}
> \frac{\left\lVert Ax \right\rVert_{b}}{\left\lVert x \right\rVert_{a}}
> $$
>
> where $\|\cdot\|_a$ and $\|\cdot\|_b$ are common vector norms. For example $\|x\|_2=\sqrt{\sum_ix_i^2}$ and $\|x\|_1=\sum_i |x_i|$, etc. We will include this definition in the updated draft.
>
> **Weakness 5**
>
> > 5. The notation is somewhat confusing, for instance, in line 383 of algorithm 1, $g^t_l$ is computed and it's hard to tell where it's used since it doesn't appear anywhere else in the algorithm.
>
> **Response** We thank the reviewer for pointing this out! We apologize that the algorithm looks confusing due a typo. On line 385, the update should be:
> $$
> m_l^t = \beta_tm_{l}^{t-1}+(1-\beta_t)g_l^t
> $$
> and line 387 should be:
> $$
> m_l^t = g_l^t
> $$
> So basically $g_l^t$ is used for computing the momentum $m_l^t$. We updated it accordingly in the revised draft.

---

> ### Author Response · Authors · 2025-11-22
>
> **Weakness 6:**
>
> > 6. The proof of theorem 2.1 is hard to follow since it contains a series of equations. It will be helpful to give a larger overview of the proof and highlighting the steps to prove them, and the purpose of the lemmas.
>
> **Response:**
>
> We thank the reviewer for the valuable suggestion. Lemma A.1 and A.2 together quantifies how good the momentum $m_l^t$ is approximating the true gradient $\nabla_{\theta_l}\ell(\theta^t)$. Proposition A.1 is the place where we use the Lipschitz smoothness of the loss function to quantify the process we make after updating for one iteration (whereas Lemma A.3 constructs an auxiliary sequence which help the proof of Proposition A.1). Combining Lemma A.1 and A.2 and Proposition A.1 provides the final result in Theorem 2.1.
>
> For more details we please refer to our updated draft. We have included more explanations on the meaning of each terms in Theorem 2.1 and a short proof sketch after the theorem statement.
>
> **Weakness 7:**
>
> > 7. Also, please mention an overview of the proofs of the lemmas, and how the inequalities follow from each other, since they are quite long and tedious to verify.
>
> **Response:** We thank the reviewer for the pointing this out. Please refer to the answer above for a brief explanation of the proof. Again, in the update draft, we included a more detailed proof sketch after the theorem statement.
>
> **Question 1:**
>
> > 1. Is variance of stochastic gradient computed over batches or individual training data?
>
> **Response:**
>
> We thank the reviewer for the question. To answer it directly: we compute it in batches. Below is a detailed explanation.
>
> The variance of the stochastic gradient is computed in the following sense: first, gather a batch of data (say, batchsize=64) and compute the stochastic gradient via back-propogation; then, gather another batch of data (with much larger batchsize, say 1024), and compute the "true" gradient via back-propagation again; last, compute and record the distance of these two stochastic gradients. This is a surrogate since it is impossible to load all the data into memory and compute the gradient on the entire dataset (which for LLM pretraining consists of billions of training samples). We hope this answers the reviewer's question.
>
> **Question 2:**
>
> > 2. Could you discuss the weaknesses above, especially giving proof intuitions?
>
>
> **Response:** Please refer to our responses to the above weaknesses section.

---

### Official Review · Reviewer_75sN · 2025-11-01

**Soundness:** 3
**Presentation:** 4
**Contribution:** 3
**Rating:** 4
**Confidence:** 5

**Summary:**

This paper proposes SCALE (Stochastic Column-normalized Last-layer Momentum) — a minimalist, memory-efficient optimizer for LLM pretraining. Instead of modifying Adam or introducing compression subspaces like GaLore, Fira, or APOLLO, the authors take a bottom-up approach to identify the minimal ingredients necessary for high-performance pretraining under tight memory budgets. They find that (i) column-wise gradient normalization and (ii) applying first-order momentum only to the last layer are sufficient to achieve Adam-level performance while consuming only ~35–45% of the memory.
Extensive experiments on LLaMA models (60M–7B) show that SCALE matches or surpasses Adam and state-of-the-art memory-efficient optimizers (APOLLO, Muon, SWAN) in perplexity–memory trade-offs.

**Strengths:**

* **Clarity and thoroughness**： The methodology is clearly articulated，and easy to understand.
* **Memory saving**: SCALE is lightweight, simple to implement, and achieves Adam-like performance at one-third the memory cost, even beat the SOTA APOLLO.
* **Originality**: The paper clearly present how to ablate normalization and momentum mechanisms to find the essential ingredients that make adaptive optimizers effective. This is good since whether AdamW is good for high-dim LLM optimziation should be challenged.

**Weaknesses:**

I have one major concern, which is the generality of the techniques in the paper:
* **Potential overfitting to a single model family.**
The main design choices (column-wise normalization and last-layer momentum) are validated only on LLaMA-style architectures. It remains unclear whether these techniques generalize to other architectures. This is important as the paper's contribution is building a minimal optimizer via ablation; how the conclusion can be extended would be the main concern or flaw of this paper. This is my main reason to give a negative score now, but I am happy to see more results to show the generality of the techniques.

**Questions:**

* It remains unclear whether these techniques generalize to other architectures.
* Lack of fine-tuning or SFT experiments. The results focus exclusively on pretraining. Adding SFT experioments would be more convincing.

---

> ### Author Response · Authors · 2025-11-22
>
> We thank the reviewer for the positive comments about our paper and method! Bellow we provide our detailed response to the raised concern and questions.
>
> **Weakness**
>
> > I have one major concern, which is the generality of the techniques in the paper:
>
> > Potential overfitting to a single model family. The main design choices (column-wise normalization and last-layer momentum) are validated only on LLaMA-style architectures. It remains unclear whether these techniques generalize to other architectures. This is important as the paper's contribution is building a minimal optimizer via ablation; how the conclusion can be extended would be the main concern or flaw of this paper. This is my main reason to give a negative score now, but I am happy to see more results to show the generality of the techniques.
>
> **Response:**
>
> **TL;DR**: We performed additional experiments on Qwen2, GPT2 and Gemma model architectures to verify our method's generality. Our method maintains SOTA memory-efficient petraining performance, giving Adam-like results and surpassing strong baselines such as APOLLO, both in terms of memory efficiency and resulting perplexity. Bellow is our detailed answer:
>
>
> We choose the LLaMA family of models as it is commonly used to benchmark memory efficient optimizers for LLM pretraining (such as in GaLore, Fira, SWAN, APOLLO papers), which is the scope of our paper. However, we fully agree with the reviewer that results on other LLM architectures will provide further evidence of our method's generality. Thus, we conducted pretraining experiments with Qwen2-500M  and GPT2-Medium (355M) using the experimental settings and hyperparameter search for all baselines as described in our paper, trained on 10B and 7.8B tokens, respectively. We show the resulting perplexities (lower is better) bellow:
>
> | Method / Model | Qwen2-500M | GPT2-Medium |
> |--------            |------------|--------     |
> | Adam               |  17.61 (2.96G) |    20.73   (2.13G)   |
> | Adam (Stable-SPAM) |  15.91 (2.96G) |    18.90 (2.13G)   |
> | Muon               |  16.03 (1.98G) |    19.61 (1.42G)   |
> | GaLore             |  18.22 (1.94G) |    23.66   (1.22G)   |
> | Fira               |  15.94 (1.94G) |   19.41  (1.22G)   |
> | APOLLO             |  16.04 (1.94G) |   19.30  (1.22G)   |
> | APOLLO-Mini        |  16.17 (1.53G) |   19.99  (0.92G)   |
> | SCALE              |  **15.57** (**1.26G**) |   **19.00**  (**0.81G**) |
>
> We observe that our method still achieves Adam-like performance while using significantly less memory and continues to outperform SOTA memory-efficient methods including APOLLO. We believe that our simple, yet systematic, optimizer design makes our method more robust and less likely to overfit to specific LLM architectures.
>
> Additionaly, bellow we include new pretraining results with the Gemma-2B model on 32B tokens. Due to the larger model size and the limited rebuttal time we were only able to provide a comparison with the strongest memory-efficient baseline, APOLLO. We report the resulting perplexities (lower is better) bellow:
>
> | Method / Model     |   Gemma-2B     |
> |--------            |------------    |
> | APOLLO             |  12.05     (9.09G) |
> | SCALE              |  **11.96** (**6.06G**) |
>
> We can see that our method achieves a lower perplexity with an even smaller memory footprint than APOLLO, which further showcases its SOTA performance.
>
> **Question 1:**
> > 1. It remains unclear whether these techniques generalize to other architectures.
>
> **Response:** We refer the reviewer to our response to the weakness above. We have also included our new results to the Appendix of our updated draft.

---

> > ### Author Response · Authors · 2025-11-22
> >
> > **Question 2:**
> > > 2. Lack of fine-tuning or SFT experiments. The results focus exclusively on pretraining. Adding SFT experiments would be more convincing.
> >
> > **Response:**
> >
> > We thank the reviewer for the question. We want to emphasize that our optimizer design is focused on memory-efficient LLM pretraining. As pretraining is by far the most computationally expensive part of the LLM training process (for example pretraining LLaMA2-70B took about 1.72 million A100 GPU hours [1]), it is commonly  believed that an efficient SOTA pretraining optimizer is very important.
> >
> > Instead, for fine-tuning, existing Parameter-efficient fine-tuning (PEFT) methods already reduce the memory and compute requirements substantially while giving performance competitive to full fine-tuning. Therefore we don't expect our optimizer to replace such approaches. However, to give further evidence of its generality, we provide some preliminary results on GLUE benchmarks with a pretrained RoBERTa-base which we fine-tuned with our method and Adam (full fine-tuning), following the setup of [2]. We conduct a learning rate search for both methods and report the best epoch results (higher the better).
> >
> > | Method / Benchmark | RTE | CoLA  | MRPC | STS-B | SST-2 | QNLI | Avg |
> > |--------            |------------|--------|--------|--------|--------|--------| --------|
> > |Adam    (0.75G)      | 79.53 | 63.37 | 93.01 | 91.26 | 94.26 | 92.67 | 85.68 |
> > |SCALE   (0.33G)      | 80.46 | 63.02 | 92.53 | 91.23 | 93.58 | 92.23 | 85.51 |
> >
> > We observe that our method, despite being designed for pretraining, can give results comparable to Adam for fine-tuning tasks while using over 2 times less memory (included in parenthesis). Again, we want to point out that our optimizer is aimed at memory-efficient pretraining where its advantages are most relevant. As a future work, it might be worth investigating combining SCALE with PEFT methods, although this is out of the scope of this paper.
> >
> > [1] Llama 2: Open Foundation and Fine-Tuned Chat Models, 2023.
> >
> > [2] GaLore: Memory-Efficient LLM Training by Gradient Low-Rank Projection, 2024.

---

### Official Review · Reviewer_Cu6E · 2025-11-01

**Soundness:** 3
**Presentation:** 3
**Contribution:** 3
**Rating:** 6
**Confidence:** 4

**Summary:**

This paper investigates the minimal modifications required to make SGD competitive with adaptive optimizers like Adam for large-scale LLM pretraining. Through a systematic ablation, the authors identify two key components: (i) column-wise gradient normalization and (ii) first-order momentum applied only to the last layer, and then integrate them into a new optimizer, SCALE (Stochastic Column-normalized Last-layer momEntum). SCALE matches or exceeds Adam-level performance while using only 35–45% of the memory across LLaMA models ranging from 60M to 7B parameters.

**Strengths:**

- The paper addresses a well-defined and practically important question: identifying the minimal components required for memory-efficient yet high-performing LLM training.

- The set of baselines covered is comprehensive, and the systematic investigation of essential design elements is solid and convincing.

- SCALE demonstrates an outstanding memory–perplexity trade-off, effectively establishing a new Pareto frontier in optimizer efficiency.

**Weaknesses:**

Since the proposed method mainly combines known techniques (normalization and partial momentum) rather than introducing new algorithmic concepts, it would strengthen the paper to provide a deeper analysis of the normalization component. For instance, the poor performance of row-wise normalization appears to stem from the LM head layer. If the last layer is excluded, it would be interesting to investigate whether different layers exhibit specific preferences for normalization granularity, such as normalizing along the smaller or larger dimension of the gradient tensor. Additionally, exploring block-wise normalization (e.g., aligned with attention heads?) or developing a variance-based, principled normalization criterion could yield more general insights. Such a deeper analysis would make the work more convincing and theoretically insightful.

I am still somewhat unclear about why the combination of SGD, normalization, and partial momentum can achieve performance comparable to Adam. Could the authors provide deeper insights or theoretical intuition into this behavior? Is the observed effectiveness related to properties of the data distribution or gradient statistics?

**Questions:**

Please refer to weakness

---

> ### Author Response · Authors · 2025-11-22
>
> We thank the reviewer for the kind feedback and for recognizing the effectiveness of our method. Bellow we provide detailed responses to each weakness.
>
> **Weakness 1:**
> >  Since the proposed method mainly combines known techniques (normalization and partial momentum) rather than introducing new algorithmic concepts, it would strengthen the paper to provide a deeper analysis of the normalization component. (...) Such a deeper analysis would make the work more convincing and theoretically insightful.
>
> **Response**:
>
> We very much appreciate the suggestions of the reviewer. We agree that certain layers may favor normalizing their gradients along a specific dimension. In the case of LLM pretraining, which is the focus of our paper, our experimental evidence suggests that the last layer significantly benefits from column-wise normalization as opposed to row-wise. Bellow we include new ablations we conducted, as suggested by the reviewer, to further investigate this behavior:
>
> | Method                              | Perplexity |
> |--------                             |------------|
> | 1. SCALE (all column, mmt-last)     |  **22.57**     |
> | 2. column-last, row-rest, mmt-last  |  23.27     |
> | 3. row-first, column-rest, mmt-last |  22.94     |
> | 4. norm along larger dim, mmt-last  |  23.52     |
> | 5. row-last,  column-rest, mmt-last |  28.83     |
>
> All results are from pretraining the LLaMA 130M model using momentum only for the last layer (mmt-last, as in SCALE). Methods 1-4 all do column-wise normalization for the last layer; only method 5 (row-last,  column-rest, mmt-last) does row-wise for the last layer (row-last). Note that "column-rest" or "row-rest" means we do column- or row-wise normalizations for all other layers we don't explicitly state.
>
> We can clearly observe from the above results that row-last significantly degrades the performance. In addition, we can see that excluding the last layer, the rest of the layers are less influenced by the normalization direction. However, a uniform normalization approach as in SCALE (all column-wise) still outperforms other "mixed" normalization approaches.
>
> To explain the above results, our intuition suggests that the last layer in LLMs is somehow "special". Indeed, it is the linear projection layer (often called Language modeling head) that maps the last hidden state (of dimension d_model) to the vocabulary logits (of dimension |V| being the vocabulary size). Its gradient matrix shape is d_model x |V|, i.e., it has |V| columns, same as its weight matrix shape. Therefore, each gradient column of the last layer has a "physical meaning", i.e., it corresponds to one of the |V| vocabulary tokens.
>
> We have observed that for certain (few) tokens, their corresponding last-layer column gradients have much larger norms compared to the rest. We have also observed that such columns correspond to more frequent tokens (**see our updated draft, Appendix A.10 for figures and details**). We hypothesize that such last-layer column-norm differences can potentially lead to uneven learning (i.e., more rare tokens receive smaller gradients and therefore are not "learned" by the model) or even training divergence (due to the large gradients corresponding to frequent tokens). In connection to this, [1], aiming at explaining the large Adam-SGD gap in LLM training, observed that SGD is able to make much less progress on learning low-frequency classes compared to Adam. Based on those insights, we conclude that by normalizing the columns of the last layer our method obtains more stable training dynamics and a more "even" token learning behavior, even without using full per-parameter adaptivity as in Adam.
>
> Although still at a preliminary stage, we hope that our above insights for the importance of column-wise normalization for the last layer can provide a deeper understanding of the normalization component and (at least to some degree) explain why SCALE is able to perform comparable to Adam for LLM pretraining.

---

> ### Author Response · Authors · 2025-11-22
>
> **Weakness 2:**
>
> > I am still somewhat unclear about why the combination of SGD, normalization, and partial momentum can achieve performance comparable to Adam. Could the authors provide deeper insights or theoretical intuition into this behavior? Is the observed effectiveness related to properties of the data distribution or gradient statistics?
>
> **Response**:
>
> We thank the reviewer for the question. We note that, as discussed in the paper, Adam is a moving average version of a point-wise normalization, i.e. the sign SGD (Eq. 4 and 5). Also, in Table 2 we experimentally show that column as well as singular-value based normalizations perform better than point-wise sign-normalization. In addition, recent works such as Muon [2] or Scion [3] which employ matrix-based normalization (i.e., singular value normalization) and (full) momentum and have also shown results comparable or better than Adam for LLM pretraining.
>
> Therefore intuitively, we believe it is that these matrix normalizations are more suitable for matrix weights (which are common for attention-based networks), and these matrix nomalizations already improve the performance profoundly without adding any second-order momentums. Furthermore, specifically for the column-wise normalization, we provide additional intuition for its performance in answer to the reviewer's previous question.
>
> Furthermore, Section 2.2 in our paper we further show that **first-order** momentum is not needed for most of the layers, but only for the layer which has by far the largest variance, namely the LM Head layer. This further indicates that Adam actually employs a redundant amount of momentum, which has limited benefit to the overall performance.
>
> To answer the second question mark, on the gradient statistics point of view, we have shown in Figure 3 that column-wise normalization provides a more uniform gradient value, which could help stabilize training (as we further explain in our answer to weakness 1). In terms of data distribution, as LLM training data are characterized by a heavy-tailed class imbalance [1], we hypothesize that column-wise gradient normalization (especially for the last-layer) enables our method to make progress even on less frequent tokens (see our discussion to weakness 1 above for details).
>
> [1] Kunstner, Frederik, et al. "Heavy-Tailed Class Imbalance and Why Adam Outperforms Gradient Descent on Language Models". (2024).
>
> [2] Jordan, Keller, et al. "Muon: An optimizer for hidden layers in neural networks." (2024).
>
> [3] Pethick, Thomas, et al. "Training Deep Learning Models with Norm-Constrained LMOs." 2025.

---

### Author Response · Authors · 2025-12-02
**Reviews/Responses summary (Part 1/2)**

Dear AC,

We sincerely appreciate the time and effort you devoted to reviewing our work. Due to the unfortunate circumstances of this year's conference, that resulted into the early termination of the discussion phase, **we didn't receive any response from the reviewers to our rebuttal**. In hopes of easing your increased workload, we provide a summary of the reviews and our responses. **We believe the extensive new experiments and clarifications address the reviewers' initial concerns.**

### **Reviewer Cu6E:**

Soundness: 3: good, Presentation: 3: good, Contribution: 3: good, Rating: 6.

**Strengths:** Importance of question addressed; comprehensive baselines; solid/convincing design approach; and

"*SCALE demonstrates an outstanding memory–perplexity trade-off, effectively establishing a new Pareto frontier in optimizer efficiency*."

**Weaknesses:** Request for deeper analysis of the normalization component; request for deeper insights/intuition about the effectiveness of our method.

**Rebuttal:** We included new ablations for the normalization component that validate our choice and provided new insights (Appendix A.10). We also highlighted our empirical and theoretical results regarding last-layer momentum.

### **Reviewer 75sN:**

Soundness: 3: good, Presentation: 4: excellent, Contribution: 3: good, Rating: 4.

**Strengths:** Methodological clarity, originality of our bottom-up approach; and merits of proposed algorithm:

*"SCALE is lightweight, simple to implement, and achieves Adam-like performance at one-third the memory cost, even beat the SOTA APOLLO."*

**Weaknesses:** Concern about the generality of the techniques to LLM architectures other than from LLaMA model family. The reviewer also noted: **"This is my main reason to give a negative score now, but I am happy to see more results to show the generality of the techniques."**; Also asked about fine-tuning results.

**Rebuttal:** Benchmarking on LLaMA models (60M-7B in size) is standard choice in related works (such as in GaLore, Fira, SWAN, APOLLO papers). Nevertheless, we performed **extensive new pretraining experiments on Qwen2, GPT2 and Gemma** architectures. Our method **achieves SOTA** petraining performance, both in terms of memory efficiency and resulting perplexity. We also included fine-tuning experiments with RoBERTa-base; our method, despite being designed for pretraining, can give results comparable to Adam with more than 2x less memory.


### **Reviewer agWr:**

Soundness: 3: good, Presentation: 2: fair, Contribution: 3: good, Rating: 4.

**Strengths:** Paper/results clarity; extensive comparison with baselines; simplicity and performance of proposed algorithm; well displayed analysis of the ideas.

**Weaknesses:** Techniques have been used before (novelty concern); generality of the method; math notation issues; proofs are hard to follow. The reviewer also noted: *"Could you discuss the weaknesses above, especially giving proof intuitions?"*.

**Rebuttal:**

Regarding the novelty concern, we reminded the reviewer that the purpose of our paper is not to propose new optimization techniques but to systematically investigate minimum modifications to plain SGD by following a bottom-up approach, which gave rise to SCALE, a simple yet SOTA memory-efficient optimizer.

Regarding generality, it is made clear in our paper that the focus is LLM pretraining, where the vast majority of modern LLMs follow the standard design structure we study. Nevertheless, during the revision period, we have further demonstrated the generality of our approach by showing that it still achieves SOTA performance on GPT, Qwen and Gemma architectures (beyond LLaMA).

Regarding notation/proof intuitions, we updated the draft accordingly.

---

> ### Author Response · Authors · 2025-12-02
> **Reviews/Responses summary (Part 2/2)**
>
> ### **Reviewer CaWc:**
>
> Soundness: 4: excellent, Presentation: 4: excellent, Contribution: 3: good, Rating: 8.
>
>
> **Strengths:**
>
> Thorough experimental approach: *"The authors didn’t jump straight to proposing an algorithm (...) They ran extensive ablations to dissect what matters (...) This methodical approach builds confidence that the chosen techniques are indeed the critical ones."*
>
> Simplicity of resulting optimizer: easy adaptation; easy to maintain/understand; relation to prior works but with fewer components.
>
> Empirical results: "*SCALE offers outstanding memory-perplexity trade-offs (...) Also the method’s benefits increase with model size.*"
>
> Theoretical results and understanding: *The identification of the last layer’s gradient variance as a key issue is backed by both an empirical plot (variance curves) and a theoretical argument (Theorem 2.1). This realistically explans why momentum should applied only to the last layer.*
>
> **Weaknesses:**
>
> No fundamentally new primitives introduced. The reviewer also noted: "*One could argue that the contribution is more in the clever combination and insight rather than a brand-new algorithmic concept as it still builds upon gradient normalization and momentum.*"
>
> Unclear if the approach generalizes to other domains or tasks. The reviewer also noted: *"This isn’t exactly a weakness of what’s done (the paper already has an impressive array of experiments), but it leaves an open question about generality."*
>
>
> **Rebuttal:** For the first weakness (new primitives) our response is similar to that for reviewer agWr. For the question about generality to other domains or tasks, we re-emphasize that our target domain is LLM pretraining where  memory-efficient optimizers are of great importance. In addition, we have conducted new pretraining and finetuning experiments to showcase generality, as described in our response to reviewer 75sN.

---

### Public Comment · ~Mingyi_Hong1 · 2026-03-10
**Public comment on inconsistencies in the meta-review**

We are posting this comment for the public record, to highlight some concerning irregularities of the meta-review we recived by Area Chair s7kz.

In short, the AC (i) is citing reviewers' concerns  that are inconsistent with the actual reviews we got and (ii) introduces completely new concerns that are factually incorrect, such as **claiming omissions that are clearly addressed in our paper**.

In particular, about point (i), the AC claims that "**several** reviewers raise concerns about novelty, theoretical depth, and conceptual completeness". This is inconsistent with the actual reviews, since:


- There are **no** reviewers raising concerns about our theory. Only one reviewer (with positive score) simply suggests that extra "deeper insights or theoretical intuition" could make our work even more insightful,  which we provided in our rebuttal.
- Only one reviewer (with the lowest confidence: 3) raises a novelty concern (which we addressed in our rebuttal).
- "Conceptual completeness" is a vague term and not a specific weakness, and certainly not something pointed out by any of our reviewers.


About point (ii), the AC continues by mentioning their own, completely unjustified, concerns being "**the most significant outstanding issue**" that non of the 4 reviewers ever mentioned:

> 1) "the theoretical justification for the optimizer design appears suboptimal and potentially misleading."

Our theoretical justification matches the convergence lower bound of order $\mathcal{O}(1/\sqrt{T})$, and it is not clear to us what the AC means by claiming "suboptimal". This is also evident based on the fact that none of the reviewers raised concerns regarding the suboptimality of the convergence theory.

> 2) "While the paper positions itself as identifying "minimal" modifications to SGD it ignores discussion of several established and practically important techniques that are directly relevant to the stated goal, such as Muon-style step-size scaling (Liu et al., 2025) and the use of Adam-style update for all 1D parameters (...) their omission in the analysis weakens the claim that the proposed optimizer captures the truly essential components for memory-efficient pretraining."

- First, regarding "Muon-style step-size scaling", we directly compare with Muon from (Liu et al., 2025) that uses the said scaling technique. Nevertheless, it is evident from (Liu et al., 2025) that this is an "empirical trick" used primarily so that the learning rates of Adam can be directly used for Muon. Obviously this is a technique specifically applied for Muon optimizer, and has nothing to do with our minimalist optimizer design study building upon SGD by identifying necessary components.

- Second, we clearly state in the Appendix, as well as in our rebuttal to one of the reviewers, that we also consider Adam-style update for 1D parameters, as this is a common practice among all our baselines. Therefore this is indeed a technique we are already using and not something we "omitted".


Given the inconsistent concerns of the AC, as well as the highly positive reviews we got (with first-round scores 8-6-4-4, average 5.5,  top ~10% of all submissions) which the AC themselves has acknowledged, we find the recommendation of the AC for rejection clearly unjustified and inconsistent with the review process.

---

### Meta-Review · Area_Chair_s7kz · 2026-01-02

**Summary:**

This paper studies memory-efficient optimizers for LLM pretraining through a bottom-up analysis of SGD variants, culminating in the proposed optimizer SCALE, which combines column-wise gradient normalization with momentum applied only to the last layer. Reviewers generally agree that the paper addresses an important and timely question, and they consistently praise the **extensive experimental evaluation**, the **systematic ablation strategy**, and the **strong empirical memory–perplexity trade-offs**, which often match or exceed Adam and outperform recent memory-efficient optimizers such as GaLore, Fira, APOLLO, and Muon.

At the same time, several reviewers raise concerns about **novelty**, **theoretical depth**, and **conceptual completeness**. In particular, the method largely recombines existing techniques (gradient normalization and momentum), and its contribution is primarily empirical and organizational rather than algorithmically new. Reviewers also question whether the theoretical analysis fully explains the observed empirical behavior, especially why the specific choice of last-layer momentum is essential.

### Reviewer Concerns Addressed vs. Outstanding

The rebuttal successfully addresses several reviewer concerns. In particular:

* The authors added **new ablations and analyses** clarifying the role of column-wise normalization, especially for the LM head, and provided intuitive explanations relating to token frequency and gradient scale imbalance.
* Concerns about **generality beyond LLaMA** were addressed with additional experiments on GPT-2, Qwen, and Gemma architectures, as well as limited fine-tuning results.
* Presentation and notation issues were partially improved, and additional empirical evidence strengthens confidence in the reported results.

However, **key concerns remain unresolved**, and I would like to emphasize that the most significant outstanding issue below reflects **my own reading and assessment of the paper as Area Chair**, rather than a point raised explicitly by any single reviewer.

In particular, the **theoretical justification for the optimizer design appears suboptimal and potentially misleading**. The paper’s analysis suggests that larger momentum parameters (β) are inherently beneficial, which underpins the argument for selectively applying momentum to high-variance layers. Based on my own reading, this conclusion conflicts with well-established results on SGD with momentum, which the authors based their analysis on (Liu et al., NeurIPS 2020), which show that **SGD with momentum can achieve the same asymptotic convergence rate as SGD regardless of β** (as SGD is already optimal here), under appropriate conditions and step-size schedules. As a result, the paper’s main theoretical motivation for favoring specific momentum regimes, and by extension the necessity of the proposed design, appears to rely on a **non-tight or incomplete analysis**.

In addition, while the paper positions itself as identifying “minimal” modifications to SGD, it ignores discussion of **several established and practically important techniques** that are directly relevant to the stated goal, such as Muon-style step-size scaling  (Liu et al., 2025) and the use of Adam-style update for all 1D parameters (e.g., biases and LayerNorm weights). These techniques could be critical in large-scale LLM training, and their omission in the analysis weakens the claim that the proposed optimizer captures the truly essential components for memory-efficient pretraining.

### Recommendation

**Reject**

### Rationale

While this paper presents **impressive and carefully executed experimental results**, and the empirical performance of SCALE is undeniably strong, the **central scientific claims are weakened by suboptimal theoretical analysis and incomplete positioning relative to prior work**. In particular, based on my own assessment, the paper’s theoretical argument regarding the role of momentum does not align with existing convergence results for SGD with momentum. Furthermore, the omission of other known critical optimizer components undermines the claim of identifying truly minimal and sufficient modifications to SGD.

Overall, I view this work as a **strong empirical study with useful insights**, but it does not yet meet the bar for acceptance at ICLR due to concerns about **theoretical correctness, novelty, and completeness** relative to the stated goals.

**Reviewer Concerns:**

See above.

**Reviewer Scores:**

Given the rebuttal and additional experiments, some reviewers might slightly increase their confidence in the empirical claims or generality. However, the core novelty and theoretical soundness concerns remain, and it is unlikely that reviewers with lower scores would substantially revise their overall ratings upward. Overall, the score distribution would likely remain mixed, centered around marginal acceptance.

---

### Decision · Program_Chairs · 2026-01-26

Reject